# Casting hybrid digital-analog training into hierarchical energy-based learning

**Timothy Nest**[*][♦]
timothy.nest@mila.quebec

**Maxence Ernoult**[†][♦]
maxence@rain.ai

## Abstract

Deep learning requires new approaches to combat the rising cost of training large models. The combination of energy-based analog circuits and the Equilibrium Propagation (EP) algorithm offers one compelling alternative to backpropagation (BP) for gradient-based optimization of neural nets. In this work, we introduce a hybrid framework comprising feedforward (FF) and energy-based (EB) blocks housed on digital and analog circuits. We derive a novel algorithm to compute gradients end-to-end via BP *and* EP, through FF and EB parts respectively, enabling EP to be applied to much more flexible and realistic architectures as analog units are incorporated into digital circuitry over time. We demonstrate the effectiveness of the proposed approach, showing that a standard Hopfield Network can be split into any shape while maintaining automatic differentiation performance. We apply it to ImageNet32 where we establish new SOTA in the EP and BP-alternative literature (46% top-1). An extended version of this paper and code is available here.

## 1 Introduction

Deep learning today relies on three factors: i) GPUs, ii) feedforward (FF) models and iii) backprop (BP). With skyrocketing demands of AI compute, exploration of new compute paradigms has become an economic, and environmental necessity [Strubell et al., 2020]. One path towards this goal is analog in-memory computing [Sebastian et al., 2020], promising constant time complexity as well as reduced energy consumption [Cosemans et al., 2019]. By mapping a neural network onto a "self-learning" *energy-based* (EB) analog circuit [Kendall et al., 2020, Stern et al., 2023, Dillavou et al., 2023, Scellier, 2024] loss gradients can be derived via two physical relaxations to equilibrium [Scellier et al., 2024]. Equilibrium propagation (EP) [Scellier and Bengio, 2017], is a suitable algorithm for learning in such a setting due to its strong theoretical guarantees, scalability (among BP-alternatives) [Laborieux and Zenke, 2022, 2023] and putative( $10,000\times$ ) increase in energy-efficiency and speed [Yi et al., 2023]. Still, end-to-end EP-training presents significant challenges when executed on analog hardware, including non-ideal physical behaviors affecting both inference [Wang et al., 2023, Ambrogio et al., 2023] and parameter optimization [Nandakumar et al., 2020, Spoon et al., 2021, Lammie et al., 2024], as well as incompatibility with operations including nonlinearities, batchnorm, and attention [Spoon et al., 2021, Jain et al., 2022, Liu et al., 2023, Li et al., 2023]. One possibility is hybrid systems incorporating both FF and EB components. Indeed, the design of inferential engines made of analog and digital parts is nearing commercial maturity [Ambrogio et al., 2023], yet *in-situ* learning of such systems remains unexplored.

Here, we propose a theoretical framework to extend end-to-end gradient computation to a setting where the system may or may not be *fully* analog. Our work contends that a combination of digital

---

[*]Montreal Institute of Learning Algorithms (MILA)

[†]Rain AI

[♦]Equal contribution

Second Workshop on Machine Learning with New Compute Paradigms at NeurIPS 2024(MLNCP 2024).

*and* analog hardware, with FF *and* EB parts trained via BP *and* EP respectively, can leverage advances from both digital and analog hardware in the near-term. Specifically, we introduce *Feedforward-tied Energy-based Models* (ff-EBMs, Section 3.1) whose inference pathway is a composition of FF and EB modules (Alg. 1). We show that gradients in ff-EBMs can be computed in an end-to-end fashion (Section 3.3), via BP in FF blocks and EP in EB blocks (Theorem 3.1, Alg. 2) (Section 3.2), and that when each analog block comprises a single layer, the ff-EBM is purely FF (Lemma A.1), whose training is equivalent to BP (Corollary A.1). Finally, we demonstrate the effectiveness of our algorithm on ff-EBMs where EBM blocks are DHNs (Section 4). We show that i) gradient estimates computed by our algorithm (Alg. 2) near perfectly match gradients computed by end-to-end automatic differentiation (Fig. 2), ii) a standard DHN model can be split into a ff-EBM with equivalent layers and architecture without compromising performance and remaining on par with automatic differentiation, *with significant reductions in run-time* due to the use of smaller EB blocks (Section 4.2), iii) the proposed approach scales, yielding 46 % top-1 (70% top-5) validation accuracy on ImageNet32, beating the current SOTA for BP alternatives by a large margin.

## 2   Background

**Notations.**   Given a differentiable mapping $A : \mathbb{R}^n \to \mathbb{R}^m$, we denote its *total* derivative wrt $s_j$ as $d_{s_j}A(s) := dA(s)/ds_j \in \mathbb{R}^m$, its *partial* derivative wrt $s_j$ as $\partial_j A(s) := \partial A(s)/\partial s_j \in \mathbb{R}^m$. When $A$ takes scalar values ($m = 1$), its *gradient* wrt $s_j$ is denoted as $\nabla_j A(s) := \partial_j A(s)^\top$.

### 2.1   Energy-based models (EBMs)

For a given static input and set of weights, EBMs implicitly yield a prediction through the minimization of an energy function, making them a kind of implicit model. Namely, an EBM is defined by a (scalar) energy function $E : s, \theta, x \to E(s, \theta, x) \in \mathbb{R}$ where $x$, $s$, and $\theta$ respectively denote a static input, hidden and output neuron states, and model parameters (weights). Each such tuple defines a configuration with an associated scalar energy value. Among all configurations for an input $x$ and model parameters $\theta$, the model prediction $s_\star$ is an equilibrium state which minimizes the energy :

$$s_\star := \arg \min_s E(s, \theta, x). \tag{1}$$

### 2.2   Standard bilevel optimization

Assuming that $\nabla_s^2 E(x, s_\star, \theta)$ is invertible, note that the equilibrium state $s_\star$ implicitly depends on $x$ and $\theta$ via the implicit function theorem [Dontchev et al., 2009]. Thus our goal when training an EBM is to adjust parameters $\theta$ such that $s_\star(x, \theta)$ minimizes a cost function $\ell : s, y \to \ell(s, y) \in \mathbb{R}$ where $y$ is some ground-truth associated with $x$. More formally, our objective can be cast as a *bilevel optimization problem* [Zucchet and Sacramento, 2022]:

$$\min_\theta \mathcal{C}(x, \theta, y) := \ell(s_\star, y) \quad \text{s.t.} \quad s_\star = \arg \min_s E(s, \theta, x). \tag{2}$$

To solve Eq. (2) we compute the gradient of its outer objective $\mathcal{C}(x, \theta, y)$ wrt to $\theta$ ($d_\theta \mathcal{C}(x, \theta, y)$) and perform gradient descent over $\theta$.

### 2.3   Equilibrium Propagation (EP)

An algorithm used to train an EBM as Eq. (2) may be called an EBL algorithm [Scellier et al., 2024]. EP [Scellier and Bengio, 2017] is an EBL algorithm which computes an estimate of $d_\theta \mathcal{C}(x, \theta, y)$ in at least two phases. In its first phase, the model evolves freely to $s_\star = \arg \min_s E(s, \theta, x)$. Then, the model is slightly nudged towards decreasing values of cost $\ell$ and evolves to an equilibrium state $s_\beta$. In practice, we augment the energy function $E$ by a term $\beta \ell(s, y)$ where $\beta \in \mathbb{R}^\star$ is a *nudging factor*. Weights are updated to increase the energy of $s_\star$ and decrease that of $s_\beta$, thereby "contrasting" these two states. More formally, Scellier and Bengio [2017] prescribe in the seminal EP paper:

$$s_\beta := \arg \min_s \left[ E(s, \theta, x) + \beta \ell(s, y) \right], \quad \Delta \theta^{\mathrm{EP}} := \frac{\alpha}{\beta} \left( \nabla_2 E(s_\star, \theta, x) - \nabla_2 E(s_\beta, \theta, x) \right), \tag{3}$$

where $\alpha$ denotes some learning rate. EP comes in different flavors depending on the form of $\beta$ inside Eq. (3). *Centered* EP (C-EP), where two nudged states of opposite nudging strengths ($\pm \beta$) are contrasted, performs best in practice [Laborieux et al., 2021, Scellier et al., 2024] and reads:

$$\Delta\theta^{\mathrm{C-EP}} := \frac{\alpha}{2\beta}\left(\nabla_2 E(s_{-\beta}, \theta, x) - \nabla_2 E(s_\beta, \theta, x)\right), \tag{4}$$

# 3 Tying energy-based models with ff blocks

Here we introduce our model, its associated optimization problem and learning algorithm. We show how learning amounts to solving a multi-level optimization problem (Section 3.2), and propose a BP-EP gradient chaining algorithm as a solution (Section 3.3, Theorem 3.1, Alg. 2). We highlight that ff-EBMs reduce to standard ff nets (Lemma A.1) and the proposed BP-EP gradient chaining algorithm to standard BP (Corollary A.1) when each EB block comprises a single hidden layer. Finally, we highlight in red and blue the parts of the model and associated algorithms performed inside feedforward (digital) and EB (analog) blocks respectively.

## 3.1 Feedforward-tied Energy-based Models (ff-EBMs)

**Inference procedure.** We define *Feedforward-tied Energy-based Models* (ff-EBMs) as compositions of feedforward and EB transformations. Namely, an data sample $x$ is fed into the first FF transformation $F^1$ parameterized by some weights $\omega^1$, yielding an output $x_\star^1$. $x_\star^1$ is fed as a static input into the first EB block $E^1$ with parameters $\theta^1$, and relaxes to equilibrium $s_\star^1$. $s_\star^1$ is fed into the next FF transformation $F^1$ with weights $\omega^1$ and so on until reaching the output layer $\hat{o}$.

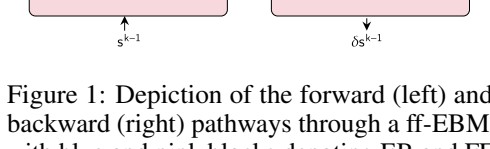

---

**Algorithm 1** ff-EBM inference

---
1: $s \leftarrow x$
2: **for** $k = 1 \cdots N - 1$ **do**
3: $\quad x \leftarrow F^k\left(s, \omega^k\right)$
4: $\quad s \leftarrow \underset{s}{\mathbf{Optim}}\left[E^k(s, \theta^k, x)\right]$
5: **end for**
6: $\hat{o} \leftarrow F^N(s, \omega^N)$

---

Figure 1: Depiction of the forward (left) and backward (right) pathways through a ff-EBM, with blue and pink blocks denoting EB and FF transformations.

A formal definition of the inference procedure for ff-EBMs is given inside Definition A.1, and more compactly inside Fig. 1 and Alg. 1.

**Form of the energy functions.** Given the $k^{\mathrm{th}}$ EB block of a ff-EBM, the associated energy function $E^k$ takes some static input $x^k$ from the output of the preceding FF transformation, has hidden neurons $s^k$ and is parameterized by weights $\theta^k$. More precisely:

$$E^k(s^k, \theta^k, x^k) := G^k(s^k) - {s^k}^\top \cdot x^k + U^k(s^k, \theta^k) \tag{5}$$

Eq. (5) reveals three contributions to the energy. The first corresponds to the non-linearity applied inside the EB block, the second to a purely FF contribution from previous FF block $F^k$, and the third to *internal* interactions within the layers of the EB block. An interesting edge case is when $U^k = 0$ for all $k$'s, i.e. no intra-block layer interactions, i.e. the EB block comprises a single layer. In this case, $s_\star^k$ is simply a feedforward mapping $x^k$ through $\sigma$ and in turn the ff-EBM is simply a standard feedforward architecture (see Lemma A.1 inside Appendix A.1.1.

## 3.2 Multi-level optimization of ff-EBMs

In the same way that learning EBMs can be cast as a bilevel optimization problem, learning ff-EBMs is a *multi-level* optimization problem where variables optimized in the inner subproblems are EB block variables $s^1, \cdots, s^{N-1}$. To make this clear, we re-write the energy function of the $k^{\mathrm{th}}$ block $E^k$ from Eq. (5) to highlight the dependence between two consecutive EB block states. Namely, by writing $\widetilde{E}^k(s^k, \theta^k, s_\star^{k-1}, \omega^k) := E^k\left(s^k, \theta^k, F^k\left(s_\star^{k-1}, \omega^{k-1}\right)\right)$, it can be seen that the equilibrium state $s_\star^k$ obtained by minimizing $E^k$ will be dependent upon the equilibrium state $s_\star^{k-1}$ of the previous

EB block, which propagates back through prior EB blocks. Thus, the learning problem for a ff-EBM [3]:

$$\min_{\theta} \ell(s_\star^{N-1}, y) \quad \text{s.t.} \quad s_\star^{N-1} = \arg\min_{s} \widetilde{E}^{N-1}(s, s_\star^{N-2}), \cdots, \text{s.t. } s_\star^1 = \arg\min_{s} \widetilde{E}^1(s, x) \quad (6)$$

### 3.3 A BP–EP gradient chaining algorithm

**Main result: explicit BP-EP chaining.** Based on the multilevel optimization problem in Eq. (6), we state the main theoretical result of this paper in Theorem 3.1 (see proof in Appendix A.2.1) and the resulting algorithm (Alg. 2), which also comes in another flavor (see Appendix A.3.1).

**Theorem 3.1** (Informal). *Assuming a ff-EBM model, we denote $s_\star^1, x_\star^1, \cdots, s_\star^{N-1}, \hat{o}_\star$ the states computed during the forward pass as depicted in Alg. 1. We define the nudged state of block $k$, denoted as $s_\beta^k$, implicitly through $\nabla_1 \mathcal{F}^k(s_\beta^k, \theta^k, x_\star^k, \delta s^k, \beta) = 0$ with:*

$$\mathcal{F}^k(s^k, \theta^k, x_\star^k, \delta s^k, \beta) := E^k(s^k, \theta^k, x^k) + \beta s^{k\top} \cdot \delta s^k \quad (7)$$

*Denoting $\delta s^k$ and $\Delta x^k$ the error signals computed at the input of the feedforward block $F^k$ and of the EB block $E^k$ respectively, then the following chain rule applies:*

$$\delta s^{N-1} := \nabla_{s^{N-1}} \ell(\hat{o}_\star, y), \quad g_{\omega^N} = \nabla_{\omega^N} \ell(\hat{o}_\star, y) \quad (8)$$

$$\forall k = 2 \cdots N-1:$$

$$\begin{cases} \Delta x^k = d_\beta \left( \nabla_3 E^k(s_\beta^k, \theta^k, x_\star^k) \right)\Big|_{\beta=0}, \quad g_{\theta^k} = d_\beta \left( \nabla_2 E^k(s_\beta^k, \theta^k, x_\star^k) \right)\Big|_{\beta=0} \\ \delta s^{k-1} = \partial_1 F^k \left( s_\star^{k-1}, \omega^k \right)^\top \cdot \Delta x^k, \quad g_{\omega^k} = \partial_2 F^k \left( s_\star^{k-1}, \omega^k \right)^\top \cdot \Delta x^k \end{cases} \quad (9)$$

---

**Algorithm 2** Implicit BP-EP gradient chaining (Theorem (3.1))

---

1: $\delta s, g_{\omega^N} \leftarrow \nabla_{s^{N-1}} \ell(\hat{o}_\star, y), \nabla_{\omega^N} \ell(\hat{o}_\star, y)$        ▷ Single backprop step
2: **for** $k = N-1 \cdots 1$ **do**
3:      $s_\beta \leftarrow \underset{s}{\textbf{Optim}} \left[ \widetilde{E}^k(s, \theta^k, s_\star^{k-1}, \omega^k) + \beta s^\top \cdot \delta s \right]$        ▷ EP through $\widetilde{E}^k$
4:      $s_{-\beta} \leftarrow \underset{s}{\textbf{Optim}} \left[ \widetilde{E}^k(s, \theta^k, s_\star^{k-1}, \omega^k) - \beta s^\top \cdot \delta s \right]$
5:      $g_{\theta^k} \leftarrow \frac{1}{2\beta} \left( \nabla_2 \widetilde{E}^k(s_\beta, \theta^k, s_\star^{k-1}, \omega^k) - \nabla_2 \widetilde{E}^k(s_{-\beta}, \theta^k, s_\star^{k-1}, \omega^k) \right)$
6:      $g_{\omega^k} \leftarrow \frac{1}{2\beta} \left( \nabla_4 \widetilde{E}^k(s_\beta, \theta^k, s_\star^{k-1}, \omega^k) - \nabla_4 \widetilde{E}^k(s_{-\beta}, \theta^k, s_\star^{k-1}, \omega^k) \right)$        ▷ i-BP through $F^k$
7:      $\delta s \leftarrow \frac{1}{2\beta} \left( \nabla_3 \widetilde{E}^k(s_\beta, \theta^k, s_\star^{k-1}, \omega^k) - \nabla_3 \widetilde{E}^k(s_{-\beta}, \theta^k, s_\star^{k-1}, \omega^k) \right)$
8: **end for**

---

## 4 Experiments

In this section, we present the ff-EBMs used in our experiments (Section 4.1) and conduct a *static* gradient analysis, where we observe that gradients computed by our algorithm are near perfectly aligned with those computed by automatic differentiation (Fig. 2). We then show on CIFAR-10 that performance of ff-EBMs does not degrade on block splits of decreasing size (despite *much lower wall clock time*)(Section 4.2). Finally, we perform further ff-EBM training experiments on CIFAR-100 and ImageNet32 where we establish a new performance state-of-the-art in the EP literature (Section 4.3).

### 4.1 Setup & static gradient analysis

**Model.** Using the same notations as in Eq. (5), the ff-EBMs at use in this section are defined through:

$$\begin{cases} U_{\text{FC}}^k(s^k, \theta^k) := -\frac{1}{2} s^{k\top} \cdot \theta^k \cdot s^k, \\ U_{\text{CONV}}^k(s^k, \theta^k) := -\frac{1}{2} s^k \bullet (\theta^k \star s^k) \end{cases}, \quad \begin{cases} F_{\text{BN}}^k(s^{k-1}, \omega^k) := \text{BN} \left( \mathcal{P} \left( \omega_{\text{CONV}}^k \star s_L^{k-1} \right); \omega_\alpha^k, \omega_\beta^k \right), \\ F_{\text{ID}}^k(s^{k-1}) := s_L^{k-1} \end{cases}$$

$$(10)$$

---

[3] we omit parameter dependence of the loss and the energy functions for readability

with $\mathrm{BN}(\cdot ; \omega_\alpha^k \omega_\beta^k)$, $\mathcal{P}$ and $\star$ the batchnorm, pooling and convolution operations, $\bullet$ the generalized dot product for tensors and $s^k := \left( s_1^{k^\top}, \cdots s_L^{k^\top} \right)^\top$ the state of block $k$ comprising $L$ layers. Weight matrices $\theta^k$ are symmetric and has a sparse, block-wise structure such that each layer $s_\ell^k$ is bidirectionally connected to its neighboring layers $s_{\ell-1}^k$ and $s_{\ell+1}^k$ through connections $\theta_{\ell-1}^k$ and $\theta_\ell^{k^\top}$ respectively (see Appendix A.1.3), either with fully connected ($U_{\mathrm{FC}}^k$) and convolutional operations ($U_{\mathrm{CONV}}^k$). The non-linearity $\sigma$ applied within EB blocks is $\sigma(x) := \min\left(\max\left(\frac{x}{2}, 0\right), 1\right)$. Additional details about equilibrium computation can be found inside Appendix A.1.3.

**Gradient comparison of EP and ID on ff-EBMs.**
We conducted comparisons between gradients obtained via our algorithm and an ID baseline (implemented as *BP through time*, see Alg. 12 inside appendix) showing near perfect alignment between ID and EP weight gradients. In practice *the use of normalization in between blocks and of the GOE weight initialization are instrumental for good gradient estimation*.

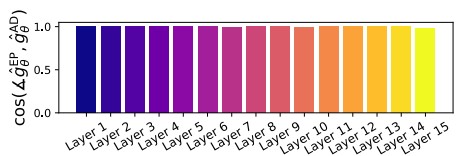

Figure 2: Cosine similarity of EP/ID grads on a random sample x,y.

## 4.2 Splitting experiment

For a given EBM and a *fixed* number layers, we considered two models of depth ($L = 6$ and $L = 12$) with various block splits maintaining equivalent architecture [4]. These experiments employ a tolerance-based (TOL) convergence. We observe that performance achieved by EP on 6-layer EBM is improved and WCT reduced with smaller blocks. Performance is maintained across 4 distinct splits between 89% and 90% and is on par with both the ID baseline for each, and with the literature [5] [Scellier et al., 2024, Laborieux and Zenke, 2022]. The same trend is seen in ff-EBMs with $L = 12$ where an accuracy of 92% across 3 splits, matching ID, and surpassing EP state-of-the art on CIFAR-10 [Scellier et al., 2024]. The significant reduction in WCT observed is due to the fact that by design training time for ff-EBMs scales linearly with number of blocks rather than supralinearly with number of layers.

| | EP | | ID | | | EP | | ID | |
|---|---|---|---|---|---|---|---|---|---|
| | Top-1 (%) | WCT | Top-1 (%) | WCT | | Top-1 (%) | WCT | Top-1 (%) | WCT |
| **L =6** | | | | | **L =12** | | | | |
| bs=6 | $89.2^{\pm 0.2}$ | 7:01 | $87.3^{\pm 0.4}$ | 6:51 | bs=4 | $89.4^{\pm 0.7}$ | 11:59 | $89.5^{\pm 0.2}$ | 8:28 |
| bs=3 | $89.8^{\pm 0.2}$ | 5:17 | $89.3^{\pm 0.2}$ | 5:10 | bs=3 | $92.5^{\pm 0.2}$ | 7:33 | $92.0^{\pm 0.1}$ | 4:16 |
| bs=2 | $90.1^{\pm 0.1}$ | 3:57 | $90.,^{\pm 0.1}$ | 4:05 | bs=2 | $92.0^{\pm 0.2}$ | 3:14 | $91.5^{\pm 0.2}$ | 3:07 |

Figure 3: Validation accuracy and Wall Clock Time (WCT) obtained on CIFAR-10 by EP (Alg. 2) and ID on models with different number of layers ($L$) and block sizes ("bs"). 3 seeds are used.

## 4.3 Scaling experiment

We consider ff-EBMs of fixed block size 2 and train them with two different depths ($L = 12$ and $L = 15$) on CIFAR-100 and ImageNet32 by EP and ID 1. We observe that EP matches ID performance on all models and tasks, and the performance obtained by training the 15-layer models by exceeds ImageNet32 SOTA for EP by 10% [Laborieux and Zenke, 2022] and by around 5% among all BP-alternatives [Høier et al., 2023].

## 5 Discussion

---

[4](e.g. for $L = 6$, 1 block of 6 layers, 2 blocks of 3 layers, etc.)
[5]in EBMs of roughly equivalent size

Table 1: Validation accuracy and WCT on CIFAR100 and ImageNet32 by EP and AD on models with different number of layers ($L$) and a BS of 2. For comparison, we show best published results for ImageNet32 by EP [Laborieux and Zenke, 2022] and all backprop alternatives [Høier et al., 2023].

| | | EP | | | ID | | |
|---|---|---|---|---|---|---|---|
| | | Top-1 (%) | Top-5 (%) | WCT | Top-1 (%) | Top-5 (%) | WCT |
| CIFAR100 | L=12 | 69.3 $^{\pm0.2}$ | 89.9 $^{\pm0.5}$ | 4:33 | 69.2$^{\pm0.1}$ | 90.0 $^{\pm0.2}$ | 4:16 |
| | L=15 | 71.2$^{\pm0.2}$ | 90.2$^{\pm1.2}$ | 2:54 | 71.1$^{\pm0.3}$ | 90.9 $^{\pm0.1}$ | 2:44 |
| ImageNet32 | L=12 | 44.7 $^{\pm0.1}$ | 61:00 $^{\pm0.1}$ | 65:23 | 44.7 $^{\pm0.6}$ | 68.9$^{\pm0.6}$ | 57:00 |
| | L=15 | **46.0** $^{\pm0.1}$ | **70.0** $^{\pm0.2}$ | 46:00 | 45.5 $^{\pm0.1}$ | 69.0 $^{\pm0.1}$ | 40:01 |
| Laborieux and Zenke [2022] | | 36.5 | 60.8 | – | – | – | – |
| Høier et al. [2023] | | 41.5 | 64.9 | – | – | – | – |

**Related work.** There is a growing body of work showing scalability of EP on vision tasks. Most notably, Laborieux and Zenke [2022] introduced holomorphic EP where loss gradients are computed with adiabatic oscillations via nudging in the complex plane Scellier et al. [2022] proposed a black-box version of EP where details about the system may not be known. These advances could be readily applied inside our EP-BP chaining algorithm to EB blocks. The work closest to ours, albeit without clear algorithmic prescriptions, is that of Zach [2021] where FF model learning is cast as a deeply nested optimization whose consecutive layers are tied by pair-wise energy functions, as does [Høier et al., 2023]. Such settings can be construed as a particular case of ff-EBM learning by EP where each EB block comprises a *single* layer ($U^k = 0$ inside Eq. (5).

**Limitations and future work.** While ff-EBM learning inherits some pitfalls of BP nothing prevents FF modules inside ff-EBMs from being trained via *any* BP alternative. BP can be parameterized by feedback weights to obviate weight transport from the inference circuit to the gradient computation circuit [Akrout et al., 2019]; its gradients approximated as finite differences of feedback operators [Ernoult et al., 2022]; or computed via implicit forward-mode differentiation with random weight perturbations in the inference circuit [Hiratani et al., 2022, Fournier et al., 2023, Malladi et al., 2023]; local layer-wise loss functions can be used to prevent "backward locking" [Belilovsky et al., 2019, Ren et al., 2022, Hinton, 2022].

One extension of this study is to incorporate *more realism into ff-EBMs*. Beyond DHNs, Deep Resistive Nets (DRNs) Scellier [2024] Kendall et al. [2020] are exact models of idealized analog circuits trainable by EP. As such, using DRNs as EB blocks inside ff-EBMs is an exciting direction, which brings new challenges [Rasch et al., 2023, Lammie et al., 2024]. Finally, considerable work is needed to prove ff-EBM further at scale on more difficult tasks (e.g. standard ImageNet), deeper architectures and novel data.

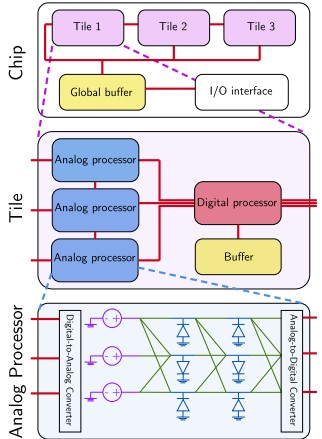

Figure 4: ff-EBMs as hierarchical systems implementing EP at chip scale (adapted from [Yi et al., 2023]) using *EB* analog processors from resistors (green ), diodes ( blue), voltage sources ( purple), ADCs and DACs (adapted from [Scellier, 2024]), digital processors, memory buffers, all linked by digital busses (red ).

**Concluding remarks and broader impact.** We show that ff-EBMs constitute a novel framework for deep-learning in heterogeneous hardware settings. We hope that this can help overcome the typical division between digital *versus* analog or BP *versus* BP-free algorithms and that the greater energy-efficiency afforded by this framework provides a pragmatic, near-term blueprint to mitigate the dramatic carbon footprint of AI training [Strubell et al., 2020].As fully analog training accelerators remain far from commercial maturity, we believe this work offers an incremental and sustainable plan for gradually integrating analog, energy-based computational primitives as they are integrated into existing digital accelerators.

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

# A  Appendix

**Contents**

## A.1 Model details

### A.1.1 Feedforward-tied EBMs (ff-EBMs)

We first formally define *Feedforward-tied Energy-based Models* (ff-EBMs) with precise assumptions on the energy-based and feedforward blocks.

**Definition A.1** (ff-EBMs). *A Feedforward-tied Energy-based Model (ff-EBM) of size $N$ comprises $N$ twice differentiable feedforward mapping $F^1, \cdots, F^N$ and $N - 1$ twice differentiable energy functions $E^1, \cdots, E^{N-1}$ with respect to all their variables. For a given $x$, the inference procedure reads as:*

$$
\begin{cases}
s^0 := x \\
x_\star^k := F^k(s_\star^{k-1}, \omega^k), \quad s_\star^k := \underset{s}{\arg\min}\, E^k(s, \theta^k, x_\star^k) \quad \forall k = 1 \cdots N - 1 \\
\hat{o}_\star := F^N(s_\star^{N-1}, \omega^N)
\end{cases}
\tag{11}
$$

*Finally, we assume that $\forall k = 1 \cdots N - 1$, $\nabla_1^2 E^k(s_\star^k, \theta^k, \omega^k)$ is invertible.*

### A.1.2 Feedforward nets as a special case

We show that when energy-based blocks comprise a single layer only, the ff-EBM becomes purely feedforward.

**Lemma A.1.** *We consider ff-EBM per Def. (A.1) where the energy functions $E^k$ have the form:*

$$
E^k(s^k, \theta^k, x^k) := G^k(s^k) - s^{k^\top} \cdot x^k + U^k(s^k, \theta^k).
\tag{12}
$$

*We assume that $U^k = 0$ for $k = 1 \cdots N - 1$, $s \to \nabla G(s)$ is invertible and we denote $\sigma := \nabla G^{-1}$. Then, the resulting model is a feedforward model described by the following recursive equations:*

$$
\begin{cases}
s_\star^0 = x \\
x_\star^k = F^k(s_\star^{k-1}, \omega^k), \quad s_\star^k = \sigma(x_\star^k) \quad \forall k = 1 \cdots N - 1 \\
\hat{o}_\star := F^N(s_\star^{N-1}, \omega^N)
\end{cases}
\tag{13}
$$

*Proof of Lemma A.1.* Let $k \in [1, N - 1]$. By definition of $s_\star^k$ and $x_\star^k$:

$$
\begin{aligned}
& \nabla_1 E^k(s_\star^k, \theta^k, x_\star^k) = 0 \\
\Leftrightarrow\quad & \nabla G^k(s_\star^k) - x_\star^k + \nabla_1 U^k(s_\star^k, \theta^k) = 0 \\
\Leftrightarrow\quad & s_\star^k = \sigma\left(x_\star^k - \nabla_1 U^k(s_\star^k, \theta^k)\right)
\end{aligned}
\tag{14}
$$

Therefore Eq. (13) is immediately obtained from Eq. (14) with $U^k = 0$.

$\square$

### A.1.3 Equilibrium computation

**For a single block.** As mentioned in Section 3.1, the energy function of the $k^{\text{th}}$ EB block has the form:

$$
E^k(s^k, \theta^k, x^k) := G^k(s^k) - s^{k^\top} \cdot x^k + U^k(s^k, \theta^k),
\tag{15}
$$

where $x^k$ is the output of the preceding feedforward block. For a given choice of a continuously invertible activation function, $G_\sigma^k$ is defined as:

$$
G_\sigma^k(s^k) := \sum_{i=1}^{\dim(s^k)} \int^{s_i} \sigma_i^{-1}(u_i) du_i \quad \text{such that} \quad \nabla G_\sigma^k(s^k)_i = \sigma_i^{-1}(s_i^k) \quad \forall i = 1 \cdots \dim(s^k).
\tag{16}
$$

To be more explicit and as we did previously, we re-write the augmented energy-function which encompasses both the $k^{\text{th}}$ EB block and the feedforward module that precedes it:

$$\widetilde{E}^k(s^k, \theta^k, s_\star^{k-1}, \omega^k) := E^k\left(s^k, \theta^k, F^k\left(s_\star^{k-1}, \omega^k\right)\right). \tag{17}$$

**Deep Hopfield Nets (DHNs) as EB blocks.** In our experiments, we used weight matrices of the form:

$$\theta^k = \begin{bmatrix} 0 & \theta_1^{k\top} & 0 & & \\ \theta_1^k & 0 & \theta_2^{k\top} & & \\ 0 & \theta_2^k & \ddots & \ddots & \\ & & \ddots & 0 & \theta_L^{k\top} \\ & & & \theta_L^k & 0 \end{bmatrix}, \tag{18}$$

whereby each layer $\ell$ is only connected to its adjacent neighbors. Therefore, fully connected and convolutional DHNs with $L$ layers have an energy function of the form:

$$U_{\text{FC}}^k(s^k, \theta^k) := -\frac{1}{2} s^{k\top} \cdot \theta^k \cdot s^k = -\frac{1}{2} \sum_\ell s_{\ell+1}^{k\top} \cdot \theta_\ell^k \cdot s_\ell^k \tag{19}$$

$$U_{\text{CONV}}^k(s^k, \theta^k) := -\frac{1}{2} s^k \bullet \left(\theta^k \star s^k\right) = -\frac{1}{2} \sum_\ell s_{\ell+1}^k \bullet \left(\theta_\ell^k \star s_\ell^k\right) \tag{20}$$

**Synchronous fixed-point iteration.** We showed that when $G$ is chosen such that $\nabla G = \sigma^{-1}$ for some activation function $\sigma$, then the steady state of the $k^{\text{th}}$ block reads:

$$s_\star^k := \sigma\left(x^k - \nabla_1 U^k(s_\star^k, \theta^k)\right), \tag{21}$$

which justifies the following fixed-point iteration scheme, when the block is influenced by some error signal $\delta s$ with nudging strength $\beta$:

$$s_{\pm\beta,t+1}^k \leftarrow \sigma\left(x^k - \nabla_1 U^k(s_{\pm\beta,t}^k, \theta^k) \mp \beta \delta s^k\right). \tag{22}$$

The dynamics prescribed by Eq. 22 are also used for the inference phase with $\beta = 0$. To further refine Eq. (22), let us re-write Eq. (22) with a layer index $\ell$ where $\ell \in [1, L_k]$ with $L_k$ being the number of layers in the $k^{\text{th}}$ block, and replacing $x^k$ by its explicit expression:

$$\forall \ell = 1 \cdots L_k : s_{\ell,\pm\beta,t+1}^k \leftarrow \sigma\left(F^k\left(s_\star^{k-1}, \omega^{k-1}\right) - \nabla_{s_\ell^k} U^k(s_{\pm\beta,t}^k, \theta^k) \mp \beta \delta s^k\right). \tag{23}$$

As done in past EP works [Ernoult et al., 2019, Laborieux et al., 2021, Laborieux and Zenke, 2022, 2023, Scellier et al., 2024] and for notational convenience, we introduce the *primitive function* of the $k^{\text{th}}$ block as:

$$\Phi^k\left(s^k, \theta^k, s_\star^{k-1}, \omega^k\right) := s^{k\top} \cdot F^k\left(s_\star^{k-1}, \omega^k\right) - U^k(s^k, \theta^k) \tag{24}$$

such that Eq. (23) re-writes:

$$\forall \ell = 1 \cdots L_k : s_{\ell,\pm\beta,t+1}^k \leftarrow \sigma\left(\nabla_{s_\ell^k} \Phi\left(s_{\pm\beta,t}^k, \theta^k, s_\star^{k-1}, \omega^k\right) \mp \beta \delta s^k\right). \tag{25}$$

Eq. (25) depicts a *synchronous* scheme where all layers are simultaneously updated at each timestep.

**Asynchronous fixed-point iteration.** Another possible scheme, employed by Scellier et al. [2024], instead prescribes to *asynchronously* update odd and even layers and was shown to speed up convergence in practice:

$$
\begin{cases}
\forall \text{ odd } \ell \in \{1, \cdots, L_k\}: & s^k_{\ell, \pm\beta, t+\frac{1}{2}} \leftarrow \sigma\left(\nabla_{s^k_\ell}\Phi\left(s^k_{\pm\beta, t}, \theta^k, s^{k-1}_\star, \omega^k\right) \mp \beta\delta s^k\right), \\
\forall \text{ even } \ell \in \{1, \cdots, L_k\}: & s^k_{\ell, \pm\beta, t+1} \leftarrow \sigma\left(\nabla_{s^k_\ell}\Phi\left(s^k_{\pm\beta, t+\frac{1}{2}}, \theta^k, s^{k-1}_\star, \omega^k\right) \mp \beta\delta s^k\right).
\end{cases}
\tag{26}
$$

We formally depict this procedure as the subroutine `Asynchronous` inside Alg. 3. In practice, we observe that it was more practical to use a *fixed* number of iterations rather than using a convergence criterion with a fixed threshold.

---

**Algorithm 3** `Asynchronous` (for all blocks **until penultimate**)

---
*Input*: $T, \theta^k, \omega^k, s^{k-1}_\star, \beta, \delta s^k$
*Output*: $s^k_\beta$
1: $s^k \leftarrow 0$
2: **for** $t = 1 \cdots T$ **do**
3: $\quad \forall \text{ odd } \ell \in \{1, \cdots, L_k\}: s^k_{\ell,\beta} \leftarrow \sigma\left(\nabla_{s^k_\ell}\Phi\left(s^k_\beta, \theta^k, s^{k-1}_\star, \omega^k\right) - \beta\delta s^k\right)$
4: $\quad \forall \text{ even } \ell \in \{1, \cdots, L_k\}: s^k_{\ell,\beta} \leftarrow \sigma\left(\nabla_{s^k_\ell}\Phi\left(s^k_\beta, \theta^k, s^{k-1}_\star, \omega^k\right) - \beta\delta s^k\right)$
5: **end for**

---

**Resulting ff-EBM inference algorithm.** With the aforementioned details in hand, we re-write the inference algorithm Alg. 1 presented in the main as a `Forward` subroutine.

---

**Algorithm 4** `Forward`

---
*Input*: $T, x, W = \{\theta^1, \omega^1, \cdots \omega^N\}$
*Output*: $s^1, \cdots, s^{N-1}$ or $\hat{o}$ depending on the context
1: $s^0 \leftarrow x$
2: **for** $k = 1 \cdots N - 1$ **do**
3: $\quad s^k \leftarrow$ `Asynchronous` $\left(T, \theta^k, \omega^k, s^{k-1}\right)$ $\qquad\qquad\qquad\qquad\qquad\qquad \triangleright$ Alg. 3
4: **end for**
5: $\hat{o} \leftarrow F^N\left(s, \omega^N\right)$

---

## A.2 Main theoretical result

### A.2.1 Proof of Theorem 3.1

The proof of Theorem 3.1 is structured as follows:

- We directly solve the multilevel problem optimization defined inside Eq. (6) using a Lagrangian-based approach (Lemma A.2), yielding optimal Lagrangian multipliers, block states and loss gradients.

- We show that by properly nudging the blocks, EP implicitly estimates the previously derived Lagrangian multipliers (Lemma A.3).

- We demonstrate Theorem 3.1 by combining Lemma A.2 and Lemma A.3.

- Finally, we highlight that when a ff-EBM is a feedforward net (Lemma A.1), then the proposed algorithm reduces to BP (Corollary A.1).

**Lemma A.2** (Lagrangian-based approach). *Assuming a ff-EBM (Def. A.1), we denote $s_\star^1, x_\star^1, \cdots, s_\star^{N-1}, \hat{o}_\star$ the states computed during the forward pass as prescribed by Eq. (11). Then, the gradients of the objective function $\mathcal{C} := \ell(\hat{o}(s_\star^{N-1}), y)$ as defined in the multilevel optimization problem (Eq. (6)), where it is assumed that $\ell$ is differentiable, read:*

$$\begin{cases} d_{\omega^N}\mathcal{C} = \partial_2 F^N(s_\star^{N-1}, \omega^N)^\top \cdot \partial_1 \ell(\hat{o}_\star, y), \\ d_{\theta^k}\mathcal{C} = \nabla_{1,2}^2 \widetilde{E}^k(s_\star^k, \theta^k, s_\star^{k-1}, \omega^k) \cdot \lambda_\star^k \quad \forall k = 1 \cdots N-1, \\ d_{\omega^k}\mathcal{C} = \nabla_{1,4}^2 \widetilde{E}^k(s_\star^k, \theta^k, s_\star^{k-1}, \omega^k) \cdot \lambda_\star^k \quad \forall k = 1 \cdots N-1, \end{cases} \tag{27}$$

*where $\lambda_\star^1, \cdots, \lambda_\star^{N-1}$ satisfy the following conditions:*

$$\begin{cases} \nabla_{s^{N-1}} \ell(\hat{o}(s_\star^{N-1}), y) + \nabla_1^2 \widetilde{E}^{N-1}(s_\star^{N-1}, \theta^{N-1}, s_\star^{N-2}, \omega^{N-1}) \cdot \lambda_\star^{N-1} = 0 \\ \forall k = N-2, \cdots, 1: \\ \quad \nabla_{1,3}^2 \widetilde{E}^{k+1}\left(s_\star^{k+1}, \theta^{k+1}, s_\star^k, \omega^{k+1}\right) \cdot \lambda_\star^{k+1} + \nabla_1^2 \widetilde{E}^k\left(s_\star^k, \theta^k, s_\star^{k-1}, \omega^k\right) \cdot \lambda_\star^k = 0 \end{cases} \tag{28}$$

*Proof of Lemma A.2.* Denoting $s := (s^1, \cdots, s^{N-1})^\top$ the state variables of the energy-based blocks, $\lambda := (\lambda^1, \cdots, \lambda^{N-1})^\top$ the Lagrangian multipliers associated with each of these variables, $W := \{\theta_1, \omega_1, \cdots, \theta_{N-1}, \omega_{N-1}\}$ the energy-based and feedforward parameters and $\hat{o}(s^{N-1}) := F^N\left(s^{N-1}, \omega^{N-1}\right)$ the logits, the Lagrangian of the multilevel optimization problem as defined in Eq. (6) reads:

$$\mathcal{L}(s, \lambda, W) := \ell\left(\hat{o}(s^{N-1}), y\right) + \sum_{k=1}^{N-1} \lambda^{k\top} \cdot \nabla_1 \widetilde{E}^k(s^k, \theta^k, s^{k-1}, \omega^k), \quad s^0 := x \tag{29}$$

Writing the associated Karush-Kuhn-Tucker (KKT) conditions $\partial_{1,2}\mathcal{L}(s_\star, \lambda_\star, W) := 0$ satisfied by optimal states and Lagrangian multipliers $s_\star$ and $\lambda_\star$, we get :

$$\nabla_1 \widetilde{E}^k(s_\star^k, \theta^k, s_\star^{k-1}, \omega^k) = 0 \quad \forall k = 1, \cdots, N-1 \tag{30}$$

$$\nabla_{s^{N-1}} \ell(\hat{o}(s_\star^{N-1}), y) + \nabla_1^2 \widetilde{E}^{N-1}(s_\star^{N-1}, \theta^{N-1}, s_\star^{N-2}, \omega^{N-1}) \cdot \lambda_\star^{N-1} = 0 \tag{31}$$

$$\nabla_{1,3}^2 \widetilde{E}^{k+1}\left(s_\star^{k+1}, \theta^{k+1}, s_\star^k, \omega^{k+1}\right) \cdot \lambda_\star^{k+1} + \nabla_1^2 \widetilde{E}^k\left(s_\star^k, \theta^k, s_\star^{k-1}, \omega^k\right) \cdot \lambda_\star^k = 0 \quad \forall k = N-2, \cdots, 1 \tag{32}$$

Eq. (30) governs the bottom-up block-wise relaxation procedure (as depicted in Alg. 1), while Eq. (31) and Eq. (32) governs error propagation in the last block and previous blocks respectively. Given $s_\star$ and $\lambda_\star$ by Eq. (30) – Eq. (32), the *total* derivative of the loss function with respect to the model parameters read:

$$d_W \ell(\hat{o}_\star, y) = d_W \left( \ell(\hat{o}_\star, y) + \sum_{k=1}^{N-1} \lambda_\star^{k\top} \cdot \underbrace{\nabla_1 \widetilde{E}^k(s_\star^k, \theta^k, s_\star^{k-1}, \omega^k)}_{=0 \quad (\text{Eq. (30)})} \right)$$

$$= d_W \mathcal{L}(s_\star, \lambda_\star, W)$$

$$= d_W s_\star^\top \cdot \underbrace{\partial_1 \mathcal{L}(s_\star, \lambda_\star, W)}_{=0 \quad (\text{Eq. (30)})} + d_W \lambda_\star^\top \cdot \underbrace{\partial_2 \mathcal{L}(s_\star, \lambda_\star, W)}_{=0 \quad (\text{Eq. (31)–(32)})} + \partial_3 \mathcal{L}(s_\star, \lambda_\star, W)$$

$$= \partial_3 \mathcal{L}(s_\star, \lambda_\star, W) \tag{33}$$

More precisely, applying Eq. (33) to the feedforward and energy-based block parameters yields:

$$d_{\omega^N} \ell(\hat{o}_\star, y) = \partial_2 F^N(s_\star^{N-1}, \omega^N)^\top \cdot \nabla_1 \ell(\hat{o}_\star, y),$$
$$d_{\theta^k} \ell(\hat{o}_\star, y) = \nabla_{1,2}^2 \widetilde{E}^k(s_\star^k, \theta^k, s_\star^{k-1}, \omega^k) \cdot \lambda_\star^k \quad \forall k = 1 \cdots N-1$$
$$d_{\omega^k} \ell(\hat{o}_\star, y) = \nabla_{1,4}^2 \widetilde{E}^k(s_\star^k, \theta^k, s_\star^{k-1}, \omega^k) \cdot \lambda_\star^k \quad \forall k = 1 \cdots N-1$$

$\square$

**Lemma A.3** (Computing Lagrangian multipliers by EP). *Under the same hypothesis as Lemma A.2, we define the nudged state of block $k$, denoted as $s_\beta^k$, implicitly through $\nabla_1 \mathcal{F}^k(s_\beta^k, \theta^k, x_\star^k, \delta s^k, \beta) = 0$ with:*

$$\mathcal{F}^k(s^k, \theta^k, x_\star^k, \delta s^k, \beta) := E^k(s^k, \theta^k, x_\star^k) + \beta s^{k\top} \cdot \delta s^k. \tag{34}$$

*Defining $(\delta s^k)_{k=1\cdots N-1}$ recursively as:*

$$\delta s^{N-1} := \nabla_{s^{N-1}} \ell(\hat{o}_\star, y), \quad \delta s^k := d_\beta \left( \nabla_3 \widetilde{E}^{k+1} \left( s_\beta^{k+1}, \theta^{k+1}, s_\star^k, \omega^{k+1} \right) \right) \Big|_{\beta=0} \quad \forall k = 1 \cdots N-2, \tag{35}$$

*then we have:*

$$\lambda_\star^k = d_\beta \left( s_\beta^k \right) |_{\beta=0} \quad \forall k = 1 \cdots N-1, \tag{36}$$

*where $(\lambda_k)_{k=1\cdots N-1}$ are the Lagrangian multipliers associated to the multilevel optimization problem defined in Eq. (6).*

*Proof of Lemma A.3.* We prove this result by backward induction on $k$.

**Initialization** ($k = N-1$). By definition, $s_\beta^{N-1}$ satisfies :

$$\beta \nabla_{s^{N-1}} \ell(\hat{o}_\star, y) + \nabla_1 \widetilde{E}^{N-1} \left( s_\beta^{N-1}, \theta^{N-1}, s_\star^{N-2}, \omega^{N-1} \right) = 0 \tag{37}$$

Differentiating Eq. (37) with respect to $\beta$ and evaluating the resulting expression at $\beta = 0$, we obtain:

$$\nabla_{s^{N-1}} \ell(\hat{o}_\star, y) + \nabla_1^2 \widetilde{E}^{N-1} \left( s_\star^{N-1}, \theta^{N-1}, s_\star^{N-2}, \omega^{N-1} \right) \cdot d_\beta s_\beta^{N-1} |_{\beta=0} = 0 \tag{38}$$

Substracting out Eq. (31) defining the Lagrangian multiplier $\lambda_\star^{N-1}$ and Eq. (38), we obtain:

$$\nabla_1^2 \widetilde{E}^{N-1} \left( s_\star^{N-1}, \theta^{N-1}, s_\star^{N-2}, \omega^{N-1} \right) \cdot \left( d_\beta s_\beta^{N-1} |_{\beta=0} - \lambda_\star^{N-1} \right) = 0 \tag{39}$$

By invertibility of $\nabla_1^2 \widetilde{E}^{N-1} \left( s_\star^{N-1}, \theta^{N-1}, s_\star^{N-2}, \omega^{N-1} \right)$, we therefore have that:

$$\lambda_\star^{N-1} = d_\beta s_\beta^{N-1} |_{\beta=0} \tag{40}$$

**Backward induction step** $(k + 1 \rightarrow k)$**.** Let us assume that $\lambda_\star^{k+1} = d_\beta s_\beta^{k+1}|_{\beta=0}$. We want to prove that $\lambda_\star^k = d_\beta s_\beta^k|_{\beta=0}$. Again, $s_\beta^{k+1}$ satisfies by definition:

$$\beta \delta s^k + \nabla_1 \widetilde{E}^k \left( s_\beta^k, \theta^k, s_\star^{k-1}, \omega^k \right) = 0, \quad \delta s^k := d_\beta \left( \nabla_3 \widetilde{E}^{k+1} \left( s_\beta^{k+1}, \theta^{k+1}, s_\star^k, \omega^{k+1} \right) \right)\Big|_{\beta=0}. \tag{41}$$

On the one hand, proceeding as for the initialization step, differentiating Eq. (41) with respect to $\beta$ and taking $\beta = 0$ yields:

$$\delta s^k + \nabla_1^2 \widetilde{E}^k(s_\star^k, \theta^k, s_\star^{k-1}, \omega^k) \cdot d_\beta s_\beta^k|_{\beta=0} = 0. \tag{42}$$

On the other hand, note that $\delta s^k$ rewrites :

$$\begin{aligned}
\delta s^k &= d_\beta \left( \nabla_3 \widetilde{E}^{k+1} \left( s_\beta^{k+1}, \theta^{k+1}, s_\star^k, \omega^{k+1} \right) \right)\Big|_{\beta=0} \\
&= \nabla_{1,3}^2 \widetilde{E}^{k+1} \left( s_\star^{k+1}, \theta^{k+1}, s_\star^k, \omega^{k+1} \right) \cdot ds_\beta^{k+1}\Big|_{\beta=0} \\
&= \nabla_{1,3}^2 \widetilde{E}^{k+1} \left( s_\star^{k+1}, \theta^{k+1}, s_\star^k, \omega^{k+1} \right) \cdot \lambda_\star^{k+1},
\end{aligned} \tag{43}$$

where we used at the last step the recursion hypothesis. Therefore combining Eq. (42) and Eq. (43), we get:

$$\nabla_{1,3}^2 \widetilde{E}^{k+1} \left( s_\star^{k+1}, \theta^{k+1}, s_\star^k, \omega^{k+1} \right) \cdot \lambda_\star^{k+1} + \nabla_1^2 \widetilde{E}^k(s_\star^k, \theta^k, s_\star^{k-1}, \omega^k) \cdot d_\beta s_\beta^k|_{\beta=0} = 0. \tag{44}$$

Finally, we substract out Eq. (32) and Eq. (44) to obtain:

$$\nabla_1^2 \widetilde{E}^k(s_\star^k, \theta^k, s_\star^{k-1}, \omega^k) \cdot \left( d_\beta s_\beta^k|_{\beta=0} - \lambda_\star^k \right) = 0. \tag{45}$$

We conclude again by invertibility of $\nabla_1^2 \widetilde{E}^k(s_\star^k, \theta^k, s_\star^{k-1}, \omega^k)$ that $\lambda_\star^k = d_\beta s_\beta^k|_{\beta=0}$.

$\square$

**Theorem A.1** (Formal)**.** *Assuming a ff-EBM model, we denote $s_\star^1, x_\star^1, \cdots, s_\star^{N-1}, \hat{o}_\star$ the states computed during the forward pass as prescribed by Alg. 1. We define the nudged state of block $k$, denoted as $s_\beta^k$, implicitly through $\nabla_1 \mathcal{F}^k(s_\beta^k, \theta^k, x_\star^k, \delta s^k, \beta) = 0$ with:*

$$\mathcal{F}^k(s^k, \theta^k, x_\star^k, \delta s^k, \beta) := E^k(s^k, \theta^k, x_\star^k) + \beta s^{k^\top} \cdot \delta s^k. \tag{46}$$

*Denoting $\delta s^k$ and $\Delta x^k$ the error signals computed at the input of the feedforward block $F^k$ and of the energy-based block $E^k$ respectively, $g_{\theta^k}$ and $g_{\omega^k}$ the gradients of the loss function:*

$$\forall k = 1, \cdots, N-1 : \; g_{\theta^k} := d_{\theta^k} \mathcal{C}, \qquad \forall k = 1 \cdots N : \; g_{\omega^k} := d_{\omega^k} \mathcal{C}, \tag{47}$$

*then the following chain rule applies:*

$$\delta s^{N-1} := \nabla_{s^{N-1}} \ell(\hat{o}_\star, y), \quad g_{\omega^N} = \partial_2 F^N \left( s_\star^{N-1}, \omega^N \right)^\top \cdot \nabla_1 \ell(\hat{o}_\star, y) \tag{48}$$

$$\forall k = 1 \cdots N-1 :$$

$$\begin{cases} \Delta x^k = d_\beta \left( \nabla_3 E^k(s_\beta^k, \theta^k, x_\star^k) \right)\Big|_{\beta=0}, & g_{\theta^k} = d_\beta \left( \nabla_2 E^k(s_\beta^k, \theta^k, x_\star^k) \right)\Big|_{\beta=0} \\ \delta s^{k-1} = \partial_1 F^k \left( s_\star^{k-1}, \omega^k \right)^\top \cdot \Delta x^k, & g_{\omega^k} = \partial_2 F^k \left( s_\star^{k-1}, \omega^k \right)^\top \cdot \Delta x^k \end{cases} \tag{49}$$

*Proof of Theorem A.1.* Combining Lemma A.2 and Lemma A.3, the following chain rule computes loss gradients correctly:

$$\delta s^{N-1} := \nabla_{s^{N-1}} \ell(\hat{o}_\star, y), \quad g_{\omega^N} = \partial_2 F^N \left(s_\star^{N-1}, \omega^N\right)^\top \cdot \nabla_1 \ell(\hat{o}_\star, y) \tag{50}$$

$$\forall k = 1 \cdots N - 1:$$

$$\begin{cases} \Delta s^{k-1} = d_\beta \left( \nabla_3 \widetilde{E}^k \left(s_\beta^k, \theta^k, s_\star^{k-1}, \omega^k\right) \right)\Big|_{\beta=0}, & g_{\theta^k} = \nabla_{1,2}^2 \widetilde{E}^k (s_\star^k, \theta^k, s_\star^{k-1}, \omega^k) \cdot d_\beta s_\beta^k|_{\beta=0} \\ g_{\omega^k} = \nabla_{1,4}^2 \widetilde{E}^k (s_\star^k, \theta^k, s^{k-1\star}, \omega^k) \cdot d_\beta s_\beta^k|_{\beta=0} \end{cases}$$
$$\tag{51}$$

Therefore to conclude the proof, we need to show that $\forall k = 1, \cdots, N - 1$:

$$d_\beta \left( \nabla_3 \widetilde{E}^k \left(s_\beta^k, \theta^k, s_\star^{k-1}, \omega^k\right) \right)\Big|_{\beta=0} = \partial_1 F^k \left(s_\star^{k-1}, \omega^k\right)^\top \cdot d_\beta \left( \nabla_3 E^k(s_\beta^k, \theta^k, x_\star^k) \right)\big|_{\beta=0} \tag{52}$$

$$\nabla_{1,2}^2 \widetilde{E}^k(s_\star^k, \theta^k, s_\star^{k-1}, \omega^k) \cdot d_\beta s_\beta^k|_{\beta=0} = d_\beta \left( \nabla_2 E^k(s_\beta^k, \theta^k, x_\star^k) \right)\big|_{\beta=0} \tag{53}$$

$$\nabla_{1,4}^2 \widetilde{E}^k(s_\star^k, \theta^k, s_\star^{k-1}, \omega^k) \cdot d_\beta s_\beta^k|_{\beta=0} = \partial_2 F^k \left(s_\star^{k-1}, \omega^k\right)^\top \cdot d_\beta \left( \nabla_3 E^k(s_\beta^k, \theta^k, x_\star^k) \right)\big|_{\beta=0} \tag{54}$$

Let $k \in [1, N - 1]$. We prove Eq. (52) as:

$$d_\beta \left( \nabla_3 \widetilde{E}^k \left(s_\beta^k, \theta^k, s_\star^{k-1}, \omega^k\right) \right)\Big|_{\beta=0} = d_\beta \left( \nabla_{s^{k-1}} E^k \left(s_\beta^k, \theta^k, F^k \left(s_\star^{k-1}, \omega^k\right)\right) \right)\big|_{\beta=0}$$

$$= \partial_1 F^k \left(s_\star^{k-1}, \omega^k\right)^\top \cdot d_\beta \left( \nabla_3 E^k(s_\beta^k, \theta^k, x_\star^k) \right)\big|_{\beta=0}$$

Eq. (53) can be obtained as:

$$\nabla_{1,2}^2 \widetilde{E}^k(s_\star^k, \theta^k, s_\star^{k-1}, \omega^k) \cdot d_\beta s_\beta^k|_{\beta=0} = d_\beta \left( \nabla_2 \widetilde{E}^k(s_\beta^k, \theta^k, s_\star^{k-1}, \omega^k) \right)\Big|_{\beta=0}$$

$$= d_\beta \left( \nabla_2 E^k(s_\beta^k, \theta^k, x_\star^k) \right)\big|_{\beta=0}$$

Finally and similarly, we have:

$$\nabla_{1,4}^2 \widetilde{E}^k(s_\star^k, \theta^k, s_\star^{k-1}, \omega^k) \cdot d_\beta s_\beta^k|_{\beta=0} = d_\beta \left( \nabla_4 \widetilde{E}^k(s_\beta^k, \theta^k, s_\star^{k-1}, \omega^k) \right)\Big|_{\beta=0}$$

$$= d_\beta \left( \nabla_{\omega^k} E^k(s_\beta^k, \theta^k, F^k \left(s_\star^{k-1}, \omega^k\right)) \right)\big|_{\beta=0}$$

$$= d_\beta \left( \partial_2 F \left(s_\star^{k-1}, \omega^k\right)^\top \cdot \nabla_3 E^k(s_\beta^k, \theta^k, x_\star^k) \right)\Big|_{\beta=0}$$

$$= \partial_2 F \left(s_\star^{k-1}, \omega^k\right)^\top \cdot d_\beta \left( \nabla_3 E^k(s_\beta^k, \theta^k, x_\star^k) \right)\big|_{\beta=0}$$

$$\square$$

### A.2.2 An alternative proof of Theorem 3.1

**An energy function for ff-EBMs?** While it is clear that the energy function of a ff-EBM is not $E = \sum_{k=1}^{N-1} \widetilde{E}^k$ (which would correspond in this case to the "single block" standard case), one may wonder if:

- ff-EBM inference (Alg. 1) can still be described as the minimization of some energy function?
- Therefore, if Theorem 3.1 can be derived by directly applying EP to this energy function?

We show below that this is indeed the case. We follow Zach [2021], denoting $s := (s^{1\top}, \cdots, s^{N-1\top})^\top$ and $W = \{W_1, \cdots, W_{N-1}\}$, by picking the following energy function:

$$
\mathcal{F}(s, W, x, \beta) := \sum_{k=1}^{N-1} \left\{ \widetilde{E}^k \left( s^k, W^k, s_\star^{k-1} \right) \right.
$$
$$
\left. + \left[ \nabla_3 \widetilde{E}^{k+1} \left( s^{k+1}, W^{k+1}, s_\star^k \right) - \nabla_3 \widetilde{E}^{k+1} \left( s_\star^{k+1}, W^{k+1}, s_\star^k \right) \right]^\top \cdot \left( s^k - s_\star^k \right) \right\}
$$
$$
+ \widetilde{E}^{N-1} \left( s^{N-1}, W^{N-1}, s_\star^{N-2} \right) + \beta \widetilde{\ell}(s^{N-1}, y, W^N), \tag{55}
$$

where we locally redefine $x$ as the concatenation of *all* block inputs, i.e. $x \leftarrow (x^\top, s_\star^{1\top}, \cdots, s_\star^{N-2\top})^\top$, and with $s_\star := (s_\star^{1\top}, \cdots, s_\star^{N-1\top})$ implicitly defined through $\nabla_1 \mathcal{F}(s_\star, W, x, \beta = 0) = 0$. In Lemma A.4, we show that the free steady state of the above energy function ($s_\star$ obtained with $\beta = 0$ inside Eq. (55)) indeed corresponds to the states computed by the ff-EBM inference scheme (Alg. 1).

**Lemma A.4.** *Let $\widetilde{E}^1, \cdots, \widetilde{E}^{N-1}$ be the block-wise energy functions of a ff-EBM defined per Def. A.1. Assume $s_\star$ implicitly defined through $\nabla_1 \mathcal{F}(s_\star, W, \beta = 0) = 0$ where $\mathcal{F}$ is defined by Eq. (55). Then:*

$$
s_\star^0 := x, \quad \forall k = 1, \cdots N - 1: \quad \nabla_1 \widetilde{E}^k(s_\star^k, W^k, s_\star^{k-1}) = 0 \tag{56}
$$

*Proof of Lemma A.4.* For $k = N - 1$, the stationarity condition $\nabla_{s^{N-1}} \mathcal{F}(s_\star, W, x, \beta)$ yields:

$$
\nabla_1 \widetilde{E}^{N-1} \left( s_\star^{N-1}, W^{N-1}, s_\star^{N-2} \right) + 0 = 0. \tag{57}
$$

Then, for any $1 \leq k < N - 1$, $\nabla_{s^k} \mathcal{F}(s_\star, W, x, \beta) = 0$ yields:

$$
\nabla_1 \widetilde{E}^k(s_\star^k, W^k, s_\star^{k-1}) + \underbrace{\left[ \nabla_3 \widetilde{E}^{k+1} \left( s_\star^{k+1}, W^{k+1}, s_\star^k \right) - \nabla_3 \widetilde{E}^{k+1} \left( s_\star^{k+1}, W^{k+1}, s_\star^k \right) \right]}_{=0} = 0 \tag{58}
$$

Eq. (57) and Eq. (57) indeed correspond to ff-EBM inference as depicted inside Alg. 1. □

**The EP fundamental Lemma.** For self-containedness of this paper, we restate the fundamental EP result below inside Lemma A.5.

**Lemma A.5** ([Scellier, 2021]). *Let $\mathcal{F}(s, W, x, \beta)$ be a twice differentiable function of the three variables $s$, $W$ and $\beta$. For fixed $W$, $x$ and $\beta$, let $s_\beta$ be a point that satisfies the stationarity condition:*

$$
\nabla_1 \mathcal{F}(s_\beta, W, x, \beta) = 0, \tag{59}
$$

*and suppose that $\nabla_1^2 \mathcal{F}(s_\beta, W, x, \beta)$ is invertible. Then, in the neighborhood of this point, we can define a continuously differentiable function $(x, W, \beta) \to s_\beta$ such that Eq. (59) holds for any $(x, W, \beta)$ in this neighborhood. Furthermore, we have the following identity:*

$$
d_W \left( \nabla_\beta \mathcal{F}(s_\beta, W, x, \beta) \right) = d_\beta \left( \nabla_2 \mathcal{F}(s_\beta, W, x, \beta) \right) \tag{60}
$$

In particular, Eq. (60) may be evaluated with $\mathcal{F} = E + \beta \ell$ at $\beta = 0$ to yield the EP learning rule, denoting $\mathcal{C} := \ell(s_\star, y)$ [Scellier and Bengio, 2017]:

$$
d_W \mathcal{C} = d_\beta \left( \nabla_2 \mathcal{F}(s_\beta, W, x, \beta) \right) |_{\beta=0} \tag{61}
$$

**Theorem 3.1 as a direct application of EP.** Now we know Eq. (55) defines a valid energy function for ff-EBMs and with Lemma A.5 in hand, we are ready to apply EP directly to this energy function. We rewrite below the block-wise free energy functions at use inside Theorem 3.1 and used in practice inside Alg. 2 to nudge a block of energy $\widetilde{E}^k$ given some top-down error signal $\delta^k$:

$$
\begin{cases}
\widetilde{\mathcal{F}}^k(s^k, W^k, s_\star^{k-1}, \delta s^k, \beta) := \widetilde{E}^k(s^k, W^k, s_\star^{k-1}) + \beta s^{k^\top} \cdot \delta s^k, \\
\delta s^k := d_\beta \left( \nabla_3 \widetilde{E}^{k+1} \left( s_\beta^k, W^{k+1}, s_\star^k \right) \right)\Big|_{\beta=0} \text{ if } k < N-1 \text{ else } \nabla_1 \widetilde{\ell}(s_\star^{N-1}, y, W^N)
\end{cases}
\tag{62}
$$

In Theorem A.2, we show that the direct application of Lemma A.5 to $\mathcal{F}$ as defined inside Eq. (55) yields the same gradient formula for each parameter $W^k$ and the same nudged block states as those prescribed by Theorem 3.1 for sufficiently small $\beta$.

**Theorem A.2** (Informal). *Let $\mathcal{F}$ be defined as in Eq. (55) satisfying the same assumptions as in Lemma A.5. For fixed $W$, $x$ and $\beta$, let $s_\beta$ satisfy the stationarity condition:*

$$
\nabla_1 \mathcal{F}(s_\beta, W, x, \beta) = 0.
$$

*Then, we have:*

$$
d_{W^k}\mathcal{C} = d_\beta \left( \nabla_2 \widetilde{E}^k(s_\beta^k, W^k, s_\star^{k-1}) \right)\Big|_{\beta=0}, \quad \nabla_1 \widetilde{\mathcal{F}}^k(s_\beta^k, W^k, s_\star^{k-1}, \delta s^k, \beta) = \mathcal{O}(\beta^2)
\tag{63}
$$

*Proof of Theorem A.2.* Lemma A.5 yielding:

$$
d_{W^k}\mathcal{C} = d_\beta \left( \nabla_{W^k} \mathcal{F}(s_\beta, W, x, \beta) \right)|_{\beta=0},
$$

proving Eq. (63) amounts to show that:

$$
d_\beta \left( \nabla_{W^k} \mathcal{F}(s_\beta, W, x, \beta) \right)|_{\beta=0} = d_\beta \left( \nabla_2 \widetilde{E}^k(s_\beta^k, W^k, s_\star^{k-1}) \right)\Big|_{\beta=0},
\tag{64}
$$

$$
\nabla_1 \widetilde{\mathcal{F}}^k(s_\beta^k, W^k, s_\star^{k-1}, \delta s^k, \beta) = \mathcal{O}(\beta^2)
\tag{65}
$$

On the one hand, we have:

$$
\nabla_{W^k} \mathcal{F}(s_\beta, W, x, \beta) = \nabla_2 \widetilde{E}^k(s_\beta^k, W^k, s_\star^{k-1})
$$
$$
+ \left( \nabla_{2,3}^2 \widetilde{E}^k(s_\beta^k, W^k, s_\star^{k-1}) - \nabla_{2,3}^2 \widetilde{E}^k(s_\star^k, W^k, s_\star^{k-1}) \right) \cdot \left( s_\beta^{k-1} - s_\star^{k-1} \right)
\tag{66}
$$

For notational convenience, we define $A(\beta) := \left( \nabla_{2,3}^2 \widetilde{E}^k(s_\beta^k, W^k, s_\star^{k-1}) - \nabla_{2,3}^2 \widetilde{E}^k(s_\star^k, W^k, s_\star^{k-1}) \right)$. Note that $A(\beta = 0) = 0$. Differentiating Eq. (66) with respect to $\beta$ and taking $\beta = 0$ yields:

$$
d_\beta \left( \nabla_{W^k} \mathcal{F}(s_\beta, W, x, \beta) \right)|_{\beta=0} = d_\beta \left( \nabla_2 \widetilde{E}^k(s_\beta^k, W^k, s_\star^{k-1}) \right)|_{\beta=0}
$$
$$
+ d_\beta A(\beta)|_{\beta=0} \cdot \underbrace{\left( s_{\beta=0}^{k-1} - s_\star^{k-1} \right)}_{=0} + \underbrace{A(0)}_{=0} \cdot \left( s_\beta^{k-1} - s_\star^{k-1} \right)
$$
$$
= d_\beta \left( \nabla_2 \widetilde{E}^k(s_\beta^k, W^k, s_\star^{k-1}) \right)|_{\beta=0},
$$

which proves Eq. (64).

On the other hand, the stationary condition $\nabla_{s^k} \mathcal{F}(s_\beta, W, x, \beta)$ on the last block ($k = N-1$) yields:

$$
\nabla_1 \widetilde{E}^{N-1}(s_\beta^{N-1}, W^{N-1}, s_\star^{N-2}) + \beta \nabla_1 \widetilde{\ell}(s_\beta^{N-1}, y, W^N) = 0
$$
$$
\Rightarrow \nabla_1 \widetilde{E}^{N-1}(s_\beta^{N-1}, W^{N-1}, s_\star^{N-2}) + \beta \nabla_1 \widetilde{\ell}(s_\star^{N-1}, y, W^N) = \mathcal{O}(\beta^2)
$$
$$
\Leftrightarrow \nabla_1 \widetilde{\mathcal{F}}^{N-1}(s_\beta^{N-1}, W^{N-1}, s_\star^{N-2}, \delta s^N, \beta) = \mathcal{O}(\beta^2).
\tag{67}
$$

For previous blocks, i.e. $k < N - 1$, we have:

$$\nabla_{s^k} \mathcal{F}(s_\beta, W, x, \beta) = 0$$

$$\Leftrightarrow \nabla_1 \tilde{E}^k(s_\beta^k, W^k, s_\star^{k-1}) + \nabla_3 \widetilde{E}^{k+1}\left(s_\beta^{k+1}, W^{k+1}, s_\star^k\right) - \nabla_3 \tilde{E}^{k+1}\left(s_\star^{k+1}, W^{k+1}, s_\star^k\right) = 0$$

$$\Rightarrow \nabla_1 \tilde{E}^k(s_\beta^k, W^k, s_\star^{k-1}) + d_\beta \left(\nabla_3 \widetilde{E}^{k+1}\left(s_\beta^{k+1}, W^{k+1}, s_\star^k\right)\right)\Big|_{\beta=0} = \mathcal{O}(\beta^2)$$

$$\Leftrightarrow \nabla_1 \widetilde{\mathcal{F}}^k(s_\beta^k, W^k, s_\star^{k-1}, \delta s^k, \beta) = \mathcal{O}(\beta^2). \tag{68}$$

Altogether, Eq. (67) and Eq. (68) finishes to prove Eq. (65). $\qquad\square$

### A.3 Resulting algorithms

#### A.3.1 Explicit BP-EP chaining

We presented in Alg. 2 a "pure" EP algorithm where the BP-EP gradient chaining is *implicit*. We show below, inside Alg. 5, an alternative implementation (equivalent in the limit $\beta \to 0$) where the use of BP through feedforward modules is *explicit* and which is the direct implementation of Theorem A.1. We also show the resulting algorithm when the ff-EBM reduces to a feedforward net (Lemma A.1) inside Alg. 7, highlight in blue the statements which differ from the general case presented inside Alg. 5.

---

**Algorithm 5** Explicit BP-EP gradient chaining (Theorem (3.1))

---

1: $\delta s, g_{\omega^N} \leftarrow \nabla_{s^{N-1}} \ell(\hat{o}_\star, y), \nabla_{\omega^N} \ell(\hat{o}_\star, y)$        $\triangleright$ Single backprop step
2: **for** $k = N - 1 \cdots 1$ **do**
3:      $s_\beta \leftarrow \mathbf{Optim}_s \left[ E^k(s, \theta^k, x_\star^k) + \beta s^\top \cdot \delta s \right]$        $\triangleright$ EP through $E^k$
4:      $s_{-\beta} \leftarrow \mathbf{Optim}_s \left[ E^k(s, \theta^k, x_\star^k) - \beta s^\top \cdot \delta s \right]$
5:      $g_{\theta^k} \leftarrow \frac{1}{2\beta} \left( \nabla_2 E^k(s_\beta, \theta^k, x_\star^k) - \nabla_2 E^k(s_{-\beta}, \theta^k, x_\star^k) \right)$
6:      $\Delta x \leftarrow \frac{1}{2\beta} \left( \nabla_3 E^k(s_\beta, \theta^k, x_\star^k) - \nabla_3 E^k(s_{-\beta}, \theta^k, x_\star^k) \right)$
7:      $g_{\omega^k} \leftarrow \partial_2 F^k \left( s_\star^{k-1}, \omega^k \right)^\top \cdot \Delta x$        $\triangleright$ Explicit BP through $F^k$
8:      $\delta s \leftarrow \partial_1 F^k \left( s_\star^{k-1}, \omega^k \right)^\top \cdot \Delta x$
9: **end for**

---

#### A.3.2 Recovering backprop through feedforward nets as a special case

**Corollary A.1.** *Under the same hypothesis as Theorem A.1 and Lemma A.1, then the following chain rule applies to compute error signals backward from the output layer:*

$$\begin{cases} \delta s^{N-1} := \nabla_{s^{N-1}} \ell(\hat{o}_\star, y), \quad g_{\omega^N} = \nabla_{\omega^N} \ell(\hat{o}_\star, y) \\ \Delta x^k = \sigma'(x_\star^k) \odot \delta s^k \\ \delta s^{k-1} = \partial_1 F^k \left( s_\star^{k-1}, \omega^k \right)^\top \cdot \Delta x^k, \quad g_{\omega^k} = \partial_2 F^k \left( s_\star^{k-1}, \omega^k \right)^\top \cdot \Delta x^k \end{cases} \quad (69)$$

*Proof of Corollary A.1.* Let $k \in [1, N - 1]$. As we can directly apply Theorem A.1 here, proving the result simply boils down to showing that:

$$\Delta x^k = \sigma'(x_\star^k) \odot \delta s^k \quad (70)$$

First, we notice that when $E^k$ is of the form of Eq. (12), then $\Delta x^k$ reads as:

$$\Delta x^k = d_\beta \left( \nabla_3 E^k(s_\beta^k, \theta^k, x_\star^k) \right) \big|_{\beta=0} = - d_\beta \left( s_\beta^k \right) \big|_{\beta=0}. \quad (71)$$

$s_\beta^k$ satisfies, by definition and when $U^k = 0$:

$$\sigma^{-1}(s_\beta^k) - x_\star^k + \beta \delta s^k = 0$$
$$\Leftrightarrow \quad s_\beta^k = \sigma \left( x_\star^k - \beta \delta s^k \right) \quad (72)$$

Combining Eq. (71) and Eq. (72) yields Eq. (70), and therefore, along with Theorem A.1, the chain-rule Eq. (69). $\qquad \square$

We showcase in Alg. 6 and Alg. 7 the resulting algorithms implicit and explicit BP-EP chaining respectively, with lines in blue highlighting differences with the general algorithm Alg. 2.

---

**Algorithm 6** Implicit BP-EP gradient chaining with $U^k = 0$

---

1: $\delta s, g_{\omega^N} \leftarrow \nabla_{s^{N-1}} \ell(\hat{o}_\star, y), \nabla_{\omega^N} \ell(\hat{o}_\star, y)$          ▷ Single backprop step

2: **for** $k = N-1 \cdots 1$ **do**

3:      $s_\beta,\ s_{-\beta} \leftarrow \sigma\left(x_\star^k - \beta \delta s^k\right), \sigma\left(x_\star^k + \beta \delta s^k\right)$          ▷ EP through $\widetilde{E}^k$

4:      $g_{\omega^k} \leftarrow \frac{1}{2\beta}\left(\nabla_4 \widetilde{E}^k(s_\beta, \theta^k, s_\star^{k-1}, \omega^k) - \nabla_4 \widetilde{E}^k(s_{-\beta}, \theta^k, s_\star^{k-1}, \omega^k)\right)$    ▷ i-BP through $F^k$

5:      $\delta s \leftarrow \frac{1}{2\beta}\left(\nabla_3 \widetilde{E}^k(s_\beta, \theta^k, s_\star^{k-1}, \omega^k) - \nabla_3 \widetilde{E}^k(s_{-\beta}, \theta^k, s_\star^{k-1}, \omega^k)\right)$

6: **end for**

---

---

**Algorithm 7** Explicit BP-EP gradient chaining with $U^k = 0$

---

1: $\delta s, g_{\omega^N} \leftarrow \nabla_{s^{N-1}} \ell(\hat{o}_\star, y), \nabla_{\omega^N} \ell(\hat{o}_\star, y)$          ▷ Single backprop step

2: **for** $k = N-1 \cdots 1$ **do**

3:      $\Delta x \leftarrow -\frac{1}{2\beta}\left(\sigma\left(x_\star^k - \beta \delta s^k\right) - \sigma\left(x_\star^k + \beta \delta s^k\right)\right)$

4:      $g_{\omega^k} \leftarrow \partial_2 F^k\left(s_\star^{k-1}, \omega^k\right)^\top \cdot \Delta x$          ▷ Explicit BP through $F^k$

5:      $\delta s \leftarrow \partial_1 F^k\left(s_\star^{k-1}, \omega^k\right)^\top \cdot \Delta x$

6: **end for**

---

### A.3.3 Detailed implementation of the implicit BP-EP chaining algorithm (Alg. 2)

**Nudging the last block.** From looking at the procedure prescribed by Theorem 3.1 and algorithms thereof (Alg. 2, Alg. 5), all the error signals used to nudge the EB blocks are *stationary*, including the top-most block where the loss error signal is fed in. Namely, the augmented energy function of the last block reads as:

$$\mathcal{F}^{N-1}(s^{N-1}, \theta^{N-1}, x_\star^{N-1}, \beta) := E^{N-1}(s^{N-1}, \theta^{N-1}, x_\star^{N-1}) + \beta s^{N-1^\top} \cdot \nabla_{s^{N-1}} \ell(\hat{o}_\star, y), \quad (73)$$

where $\hat{o}_\star := F^N\left(s_\star^{N-1}, \omega^N\right)$ is *constant*. Up to a constant, Eq. (74) uses the cost function *linearized around* $s_\star^{N-1}$ instead of the cost function itself. This is, however, in contrast with most EP implementations where the nudging force acting upon the EB block is usually *elastic*, i.e. the nudging depends on the current state of the EB block. More precisely, instead of using Eq. (73), we instead use:

$$\mathcal{F}^{N-1}(s^{N-1}, \theta^{N-1}, x_\star^{N-1}, \beta) := E^{N-1}(s^{N-1}, \theta^{N-1}, x_\star^{N-1}) + \beta \ell(F^N(s^{N-1}, \omega^N), y), \quad (74)$$

This results in the following asynchronous fixed-point dynamics for the last block:

$$\begin{cases} \forall \text{ odd } \ell \in \{1, \cdots, L_k\}: & s_{\ell, \pm\beta, t+\frac{1}{2}}^k \leftarrow \sigma\left(\nabla_{s_\ell^k} \Phi\left(s_{\pm\beta, t}^k, \theta^k, s_\star^{k-1}, \omega^k\right) \mp \beta \nabla_{s^k} \ell(s_{\pm\beta, t}^k, y)\right), \\ \forall \text{ even } \ell \in \{1, \cdots, L_k\}: & s_{\ell, \pm\beta, t+1}^k \leftarrow \sigma\left(\nabla_{s_\ell^k} \Phi\left(s_{\pm\beta, t+\frac{1}{2}}^k, \theta^k, s_\star^{k-1}, \omega^k\right) \mp \beta \nabla_{s^k} \ell(s_{\pm\beta, t}^k, y)\right). \end{cases}$$

The resulting `Asynchronous` subroutine, applying for the last block, is depicted inside Alg. 8.

**Readout.** Laborieux et al. [2021] introduced the idea of the "readout" whereby the last linear layer computing the loss logits is *not* part of the EB free block dynamics but simply "reads out" the state of the penultimate block. In all our experiments we use such a readout in combination with the cross entropy loss function. Using our formalism, our readout is simply the last feedforward transformation used inside $\ell$, namely $F^N(\cdot, \omega^N)$.

**Detailed implicit EP-BP chaining algorithm.** We provide a detailed implementation of our algorithm presented in the main (Alg. 2) in Alg. 11. As usually done for EP experiments, we always perform a "free phase" to initialize the block states (`Forward` subroutine, Alg. 4). Then, two

---

**Algorithm 8** Asynchronous (for **last** block)

---

*Input*: $T, \theta^{N-1}, \omega^{N-1}, \omega^N, s_\star^{k-1}, \beta, \ell$ (cost function), $y$
*Output*: $s_\beta^{N-1}$

1: $s^{N-1} \leftarrow 0$
2: **for** $t = 1 \cdots T$ **do**
3: $\quad \forall$ odd $\ell \in \{1, \cdots, L_N\}$:
4: $\quad\quad s_{\ell,\beta}^{N-1} \leftarrow \sigma\left(\nabla_{s_\ell^{N-1}}\Phi\left(s_\beta^{N-1}, \theta^{N-1}, s_\star^{N-2}, \omega^{N-1}\right) - \beta\nabla_{s_\ell^{N-1}}\ell(F^N(s^{N-1}, \omega^N), y)\right)$
5: $\quad \forall$ even $\ell \in \{1, \cdots, L_N\}$:
6: $\quad\quad s_{\ell,\beta}^{N-1} \leftarrow \sigma\left(\nabla_{s_\ell^{N-1}}\Phi\left(s_\beta^{N-1}, \theta^{N-1}, s_\star^{N-2}, \omega^{N-1}\right) - \beta\nabla_{s_\ell^{N-1}}\ell(F^N(s^{N-1}, \omega^N), y)\right)$
7: **end for**

---

nudged phases are applied to the last block and parameter gradients subsequently computed, as done classically (`BlockGradient` subroutine for the last block, Alg. 9), with an extra computation to compute the error current to be applied to the penultimate block ($\delta s^{N-2}$). Then, the same procedure is recursed backward through blocks (Alg. 10), until reaching first block.

---

**Algorithm 9** `BlockGradient` (for **last** block)

---

*Input*: $T, s_\star^{N-2}, \theta^{N-1}, \omega^{N-1}, \omega^N, \beta, \ell, y$
*Output*: $\delta s^{N-2}$

1: $s_\beta^{N-1} \leftarrow$ Asynchronous $\left(T, \theta^{N-1}, \omega^{N-1}, \omega^N, \beta, \ell, y\right)$ $\hfill \triangleright$ Alg. 8
2: $s_{-\beta}^{N-1} \leftarrow$ Asynchronous $\left(T, \theta^{N-1}, \omega^{N-1}, \omega^N, -\beta, \ell, y\right)$
3: $g_{\omega^N} \leftarrow \frac{1}{2}\left(\nabla_{s^{N-1}}\ell(F^N\left(s_\beta^{N-1}, \omega^N\right)) + \nabla_{s^{N-1}}\ell(F^N\left(s_{-\beta}^{N-1}, \omega^N\right))\right)$
4: $g_{\theta^{N-1}} \leftarrow \frac{1}{2\beta}\left(\nabla_2\widetilde{E}^{N-1}(s_\beta^{N-1}, \theta^{N-1}, s_\star^{N-2}, \omega^{N-1}) - \nabla_2\widetilde{E}^{N-1}(s_{-\beta}^{N-1}, \theta^{N-1}, s_\star^{N-2}, \omega^{N-2})\right)$
5: $g_{\omega^{N-1}} \leftarrow \frac{1}{2\beta}\left(\nabla_4\widetilde{E}^{N-1}(s_\beta^{N-1}, \theta^{N-1}, s_\star^{N-2}, \omega^{N-1}) - \nabla_4\widetilde{E}^{N-1}(s_{-\beta}^{N-1}, \theta^{N-1}, s_\star^{N-2}, \omega^{N-2})\right)$
6: $\delta s^{N-2} \leftarrow \frac{1}{2\beta}\left(\nabla_3\widetilde{E}^{N-1}(s_\beta^{N-1}, \theta^{N-1}, s_\star^{N-2}, \omega^{N-1}) - \nabla_3\widetilde{E}^{N-1}(s_{-\beta}^{N-1}, \theta^{N-1}, s_\star^{N-2}, \omega^{N-2})\right)$

---

**Algorithm 10** `BlockGradient` (for all blocks **until penultimate**)

---

*Input*: $T, s_\star^{k-1}, \theta^k, \omega^k, \beta, \delta s$
*Output*: $\delta s^{k-1}$

1: $s_\beta^k \leftarrow$ Asynchronous $\left(T, \theta^k, \omega^k, \beta, \delta s\right)$ $\hfill \triangleright$ Alg. 3
2: $s_{-\beta}^k \leftarrow$ Asynchronous $\left(T, \theta^k, \omega^k, -\beta, \delta s\right)$
3: $g_{\theta^k} \leftarrow \frac{1}{2\beta}\left(\nabla_2\widetilde{E}^k(s_\beta^k, \theta^k, s_\star^{k-1}, \omega^k) - \nabla_2\widetilde{E}^k(s_{-\beta}^k, \theta^k, s_\star^{k-1}, \omega^k)\right)$
4: $g_{\omega^k} \leftarrow \frac{1}{2\beta}\left(\nabla_4\widetilde{E}^k(s_\beta^k, \theta^k, s_\star^{k-1}, \omega^k) - \nabla_4\widetilde{E}^k(s_{-\beta}^k, \theta^k, s_\star^{k-1}, \omega^k)\right)$
5: $\delta s^{k-1} \leftarrow \frac{1}{2\beta}\left(\nabla_3\widetilde{E}^k(s_\beta^k, \theta^k, s_\star^{k-1}, \omega^k) - \nabla_3\widetilde{E}^k(s_{-\beta}^k, \theta^k, s_\star^{k-1}, \omega^k)\right)$

---

**Algorithm 11** Detailed implicit BP-EP gradient chaining

---

1: $s_\star^1, \cdots, s_\star^{N-1} \leftarrow$ Forward $(T_{\text{free}}, x, W)$ $\hfill \triangleright$ Alg. 4
2: $\delta s \leftarrow$ BlockGradient $\left(T_{\text{nudge}}, s_\star^{N-2}, \theta^{N-1}, \omega^{N-1}, \omega^N, \beta, \ell, y\right)$ $\hfill \triangleright$ Alg. 9
3: **for** $k = N-2 \cdots 1$ **do**
4: $\quad \delta s \leftarrow$ BlockGradient $\left(T_{\text{nudge}}, s_\star^{k-1}, \theta^k, \omega^k, \beta, \delta s\right)$ $\hfill \triangleright$ Alg. 10
5: **end for**

---

## A.4 Static gradient analysis

**Important foreword.** The whole subsection is dedicated to an important tool when developing code for EP research. While EP is agnostic to *how* the steady states are obtained – the EP theory only prescribes they are energy minimizers – they can be obtained in practice (i.e. *in simulations*) through fixed-point iteration schemes (see Appendix A.1.1). The below formally defines the computational graph spanned by these schemes and abstract them away into a transition function $K$ and defines three different techniques to compute gradients on this graph: *Automatic Differentiation* (AD, Prop. A.1), *Implicit Differentiation* (ID, Def. A.3) or *Equilibrium Propagation* (EP, Def. A.4). After defining each of these algorithms formally, we will state and demonstrate an equivalence between EP and ID (Theorem A.4) which we test numerically and relied upon for the development of our codebase.

### A.4.1 Static comparison of EP and ID on ff-EBMs

In order to study the *transient dynamics* of ID and EP, we define, with $W^k := \{\theta^k, \omega^k\}$:

$$\begin{cases} \widehat{g}_{W^k}^{\text{ID}}(t) := \sum_{k=0}^{T} d_{W^k(T-k)} \mathcal{C}(x, W^k, y), \\ \widehat{g}_{W^k}^{\text{EP}}(\beta, t) := \frac{1}{2\beta} \left( \nabla_{W^k} \widetilde{E}^k(s_{\beta,t}^k, W^k, s_\star^{k-1}) - \nabla_{W^k} \widetilde{E}^k(s_{-\beta,t}^k, W^k, s_\star^{k-1}) \right), \end{cases} \tag{75}$$

where $s_{\pm\beta,t}^k$ is computed with the nudging error current $\delta s^k$ computed with Alg. 2, and $T$ is the total number of iterations used for both ID and EP in the gradient computation phase.

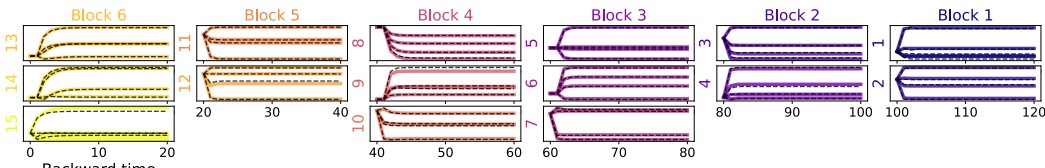

Figure 5: EP and ID partially computed gradients $((\widehat{g}_w^{\text{EP}}(t))_{t\geq 0}$ in black dotted curves and $(\widehat{g}_w^{\text{ID}}(t))_{t\geq 0}$ in plain colored curves) going *backward through equilibrium* for ID and *forward through the nudging phase* for EP [Ernoult et al., 2019] for a random sample $x$ and associated label $y$. The ff-EBM comprises 6 blocks and 15 layers in total, with block sizes of either 2 or 3 layers. Each sub-panel represents a layer (labeled on the y-axis) with each curve corresponding to a randomly selected weight. "Backward" time is indexed from $t = 0$ to $T = 120$, starting from block 6 backward to block 1, with 20 fixed-point iteration dynamics being used for both EP and ID within each block.

For a given block $k$, $d_{W^k(T-k)}\mathcal{C}(x, W, y)$ is the "sensitivity" of the loss $\mathcal{C}$ to parameter $W^k$ at timestep $T - k$ so that $\widehat{g}_{W^k}^{\text{ID}}(t)$ is a ID gradient *truncated* at $T - t$. Fig. 6 depicts the computational graph that is differentiated through when using ID and shows where $\widehat{g}_{W^k}^{\text{ID}}(t)$ are obtained correspondingly. Similarly, $\widehat{g}_{W^k}^{\text{EP}}(t)$ is an EP gradient truncated at $t$ steps forward through the nudged phase. When $T$ is sufficiently large, $\widehat{g}_{W^k}^{\text{ID}}(T)$ and $\widehat{g}_{W^k}^{\text{EP}}(T)$ converge to $d_{W^k}\mathcal{C}(x, W, y)$. Fig. 5 displays $(\widehat{g}_{W^k}^{\text{ID}}(t))_{t\geq 0}$ and $(\widehat{g}_{W^k}^{\text{EP}}(t))_{t\geq 0}$ on an heterogeneous ff-EBM of 6 blocks and 15 layers (16 if counting the last linear "readout" layer computing the logits) with blocks comprising 2 or 3 layers for a randomly selected sample $x$ and its associated label $y$ – see caption for a detailed description. It can be seen EP and ID error weight gradients qualitatively match very well throughout time, across layers and blocks. We also display the cosine similarity between the final EP and ID weight gradient estimate $\widehat{g}_{W^k}^{\text{ID}}(T)$ and $\widehat{g}_{W^k}^{\text{EP}}(T)$ for each layer and observe that EP and ID weight gradients are near perfectly aligned. Theorem A.3 generalizes the equivalence between EP and ID to ff-EBMs [Ernoult et al., 2019].

**Theorem A.3** (Informal). *Assuming* $\forall k = 1 \cdots N - 1 : s_0^k = \cdots = s_\tau^k = s_\star^k$:

$$\forall k = 1 \cdots N - 1, \forall t = 0 \cdots \tau : \quad \hat{g}_{W^k}^{\text{AD}}(t) = \hat{g}_{W^k}^{\text{ID}}(t) = \lim_{\beta \to 0} \hat{g}_{W^k}^{\text{EP}}(\beta, t) \tag{76}$$

### A.4.2 Algorithmic baselines

**Definition of the computational graph being optimized.** We abstract fixed-point iteration dynamics away into a *kernel function* $K$ which, given some block state $s_t^k$ yields $s_{t+1}^k$.

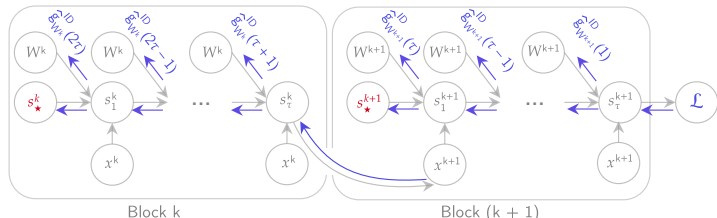

Figure 6: **Light grey:** computational graph associated with ff-EBM inference (Alg. 1) when applying fixed-point iteration to compute equilibrium states within each block (Eq. (**??**)) where the node $s_t^k$ denotes the state of block $k$ (comprising several layers) at timestep $t$. **Blue arrows:** backward automatic differentiation (AD) through the computational graph where $\hat{g}_{W^k}^{\mathrm{ID}}(t)$ is the partially computed gradient truncated at $T - t$. Since the states which are differentiated through are taken at *equilibrium* ($s_t^k = s_\star^k \ \forall t = 0 \cdots \tau$) this instantiation of AD can be viewed as *Implicit Differentiation* (ID).

**Definition A.2** (Form of the computational graph through equilibrium).

$$\forall k = 1, \cdots, N-1, \ \forall t = 1, \cdots, \tau:$$
$$x^0 = x, \quad s_t^k = K(s_{t-1}^k, W_{t-1}^k = W^k, x^k = s_\tau^{k-1}), \quad \mathcal{C} = \ell(F^N(s_\tau^{N-1}, \omega^N), y) := \tilde{\ell}(s^{N-1}, y)$$
$$(77)$$

Note that we emphasize, through the $W_{t-1}^k = W^k$ notation, that the parameters $W^k$ are shared across *all timesteps* $t = 1, \cdots, \tau$. This will help us define loss gradient with respect to $W_{t-1}^k$ further below, *i.e.* how much $W^k$ contributes *at time* $t-1$ to changing the loss $\mathcal{C}$. The *total* contribution of $W^k$ reads as the sum of the elemental contributions of all $W_t^k$. This intuition is more precisely illustrated further below. Given the computational graph defined in Def. A.2, we can now formally define the *Automatic Differentiation* (AD) baseline.

**Automatic Differentiation (AD).** Our goal is to compute:

$$g_{W^k}^{\mathrm{AD}} := \hat{g}_{W^k}^{\mathrm{AD}}(\tau) \quad \text{with:} \quad \hat{g}_{W^k}^{\mathrm{AD}}(t) := \sum_{k=1}^{t} \partial_{W_{\tau-k}^k} \mathcal{C} \tag{78}$$

In plain words, $\hat{g}_{W^k}^{\mathrm{AD}}(t)$ denotes the loss gradient for parameter $W^k$ *truncated at the $t^{\mathrm{th}}$ step* moving backward in time. We formally define below *Automatic Differentiation* (AD).

**Proposition A.1** (Automatic Differentiation (AD)). *The gradients $\hat{g}_{W^k}^{\mathrm{AD}}(t)$ can be computed using the following recursive equations:*

$$\forall k = N-1 \cdots 1:$$
$$\delta s_0^k = \delta x_\tau^{k+1} \ if \ k < N-1 \ else \ \nabla_1 \tilde{\ell}(s_\tau^{N-1}, y)$$
$$\delta x_0^k = 0, \quad \hat{g}_{W^k}^{\mathrm{AD}}(0) = 0$$
$$\forall t = 1, \cdots, \tau:$$
$$\begin{cases} \delta s_t^k = \partial_1 K(s_{\tau-t}^k, W^k, x^k = s_\tau^{k-1})^\top \cdot \delta s_{t-1}^k \\ \hat{g}_{W^k}^{\mathrm{AD}}(t) = \hat{g}_{W^k}^{\mathrm{AD}}(t-1) + \partial_2 K(s_{\tau-t}^k, W^k, x^k = s_\tau^{k-1})^\top \cdot \delta s_{t-1}^k \\ \delta x_t^k = \delta x_{t-1}^k + \partial_3 K(s_{\tau-t}^k, W^k, x^k = s_\tau^{k-1})^\top \cdot \delta s_{t-1}^k \end{cases} \tag{79}$$

*Proof of Prop. A.1.* This is a straightforward application of the chain rule applied to Eq. (77). □

**Implicit Differentiation (ID).** We define the steady state of block $k$, which we denote $s_\star^k$, as the fixed point of Eq. (77). With this notation in hand, we can define *Implicit Differentiation* (ID) in this setting.

**Definition A.3** (Implicit Differentiation (ID)). *Denoting $s_\star^k$ the fixed point of Eq. (77) inside block $k$, we define Implicit Differentiation (ID) through the following recursive equations:*

$$\forall k = N - 1 \cdots 1 :$$
$$\delta s_0^k = \delta x_\tau^{k+1} \text{ if } k < N - 1 \text{ else } \nabla_1 \tilde{\ell}(s_\tau^{N-1}, y)$$
$$\delta x_0^k = 0, \quad \hat{g}_{W^k}^{\text{ID}}(0) = 0$$
$$\forall t = 1, \cdots, \tau :$$
$$\begin{cases} \delta s_t^k = \partial_1 K(s_\star^k, W^k, x^k = s_\star^{k-1})^\top \cdot \delta s_{t-1}^{k-1} \\ \hat{g}_{W^k}^{\text{ID}}(t) = \hat{g}_{W^k}^{\text{ID}}(t-1) + \partial_2 K(s_\star^k, W^k, x^k = s_\star^{k-1})^\top \cdot \delta s_{t-1}^k \\ \delta x_t^k = \delta x_{t-1}^k + \partial_3 K(s_\star^k, W^k, x^k = s_\star^{k-1})^\top \cdot \delta s_{t-1}^k \end{cases} \quad (80)$$

We are now ready to state a simple algorithmic equivalence between ID and AD, which we built upon for our implementation of Alg. 12.

**Corollary A.2** (Equivalence of ID and AD). *Assuming that:*

$$\forall k = 1, \cdots, N - 1, \ \forall t = 1, \cdots, \tau : \quad s_t^k = s_\star^k, \quad (81)$$

*where $s_\star^k$ denotes the fixed-point of Eq. 77, then automatic differentiation (Prop. A.1) and implicit differentiation (Def. A.3) are equivalent, namely:*

$$\forall k = 1, \cdots, N - 1, \ \forall t = 1, \cdots, \tau : \quad \hat{g}_{W^k}^{\text{ID}}(t) = \hat{g}_{W^k}^{\text{AD}}(t) \quad (82)$$

*Proof of Corollary A.2.* This is a straightforward application of the definition of AD (Prop. A.1 along with the hypothesis made inside Corollary A.2. $\square$

**Resulting implementation of ID.** We describe our implementation of ID inside Alg. 12. First, we relax all blocks sequentially to equilibrium following Alg. 4 and we do not track gradients throughout this first phase, using $T_{\text{free}}$ fixed-point iteration steps per block. *Then*, initializing the block states with those computed at the previous step, we re-execute the same procedure (still with Alg. 4), this time *tracking gradients* and using $T_{\text{nudge}}$ steps fixed-point iteration steps for each block. Then, we use automatic differentiation to backpropagate through the last $T_{\text{nudge}}$ steps for each block, namely backpropagating, backward in time, *through equilibrium*.

---

**Algorithm 12** Our implementation of ID

---

1: Without tracking gradients:        ▷ e.g. with `torch.no_grad()`
2:    $s_\star^1, \cdots, s_\star^{N-1} \leftarrow \texttt{Forward}(T_{\text{free}}, x, W)$        ▷ Alg. 4
3: Initialize block states at $s_\star^1, \cdots, s_\star^{N-1}$
4: $\hat{o}_\star \leftarrow \texttt{Forward}(T_{\text{nudge}}, x, W)$        ▷ This time gradients are tracked
5: $\mathcal{C} \leftarrow \ell(\hat{o}_\star, y)$
6: Backpropagate $\mathcal{C}$ backward through the last $T_{\text{nudge}}$ steps for each block    ▷ e.g. `C.backward()`

---

**An important note about this implementation of ID.** Note that this is *not* a standard implementation of ID and it may be surprising at first glance to implement ID as AD, thereby loosing the constant $\mathcal{O}(1)$ memory cost of ID with respect to the length of the computational graph. Instead, the memory cost of Alg. 12 is $\mathcal{O}((N-1)\tau)$ [6]. However, our goal is not so much to optimize for memory usage (as in the context of Deep Equilibrium Models [Bai et al., 2019]) but to code an algorithmic baseline which we know to be equivalent to EP. Lastly, note that this implementation of ID is also known as *Recurrent Backprop* (RBP, [Almeida, 1987, Pineda, 1987]) or *Von-Neumann* RBP [Liao et al., 2018], and that ID generally comes in many more algorithmic flavors [**?**].

---

[6] We are not accounting for the *spatial depth* ($L$) of the computational graph in this cost. In this case, standard ID would have memory cost $\mathcal{O}(L)$ and our implementation inside Alg. 12 $\mathcal{O}(L(N-1)\tau)$.

### A.4.3 Extending [Ernoult et al., 2019]

In order to state a formal equivalence between EP and ID, we first need to formally define EP in the context of the aforementioned computational graph defined in Def. A.2.

**Definition A.4** (Equilibrium Propagation (EP)). *Denoting $s_\star^k$ the fixed point of Eq. (77) inside block $k$ and assuming that the transition kernel $K$ has the form $K(s, W^k, x^k) = \nabla_1 \Phi(s, W^k, x^k)$, we define Equilibrium Propagation (EP) through the following recursive equations:*

$$\forall k = N-1 \cdots 1:$$
$$\delta s^k = \Delta x_\tau^{k+1} \text{ if } k < N-1 \text{ else } \nabla_1 \tilde{\ell}(s_\tau^{N-1}, y)$$
$$\Delta x_0^k = 0, \quad \hat{g}_{W^k}^{\mathrm{EP}}(0) = 0, \quad s_{\beta,t=0}^k = s_\star^k$$
$$\forall t = 1, \cdots, \tau:$$
$$\begin{cases} s_{\beta,t+1}^k = \nabla_1 \Phi(s_{\beta,t}^k, W^k, x^k = s_\star^{k-1}) - \beta \delta s^k \\ \hat{g}_{W^k}^{\mathrm{EP}}(\beta, t) = -\frac{1}{2\beta} \left( \nabla_2 \Phi(s_{\beta,t+1}^k, W^k, x^k = s_\star^{k-1}) - \nabla_2 \Phi(s_{-\beta,t+1}^k, W^k, x^k = s_\star^{k-1}) \right) \\ \Delta x_{\beta,t}^k = -\frac{1}{2\beta} \left( \nabla_3 \Phi(s_{\beta,t+1}^k, W^k, x^k = s_\star^{k-1}) - \nabla_3 \Phi(s_{-\beta,t+1}^k, W^k, x^k = s_\star^{k-1}) \right) \end{cases}$$

Now that we have properly defined ID and EP, we are ready to state the main result of this section about the algorithmic equivalence between ID and EP which our coding work significantly built upon.

**Theorem A.4** (Extension of [Ernoult et al., 2019] to ff-EBMs). *Assuming that:*

$$\forall k = 1, \cdots, N-1, \forall t = 1, \cdots, \tau: \quad s_t^k = s_\star^k, \tag{83}$$

*where $s_\star^k$ denotes the fixed-point of Eq. (77) and that the transition kernel $K$ has the form $K(s, W^k, x^k) = \nabla_1 \Phi(s, W^k, x^k)$, then implicit differentiation (Def. A.3) and equilibrium propagation (Def. A.4) are equivalent in the limit $\beta \to 0$, namely:*

$$\forall k = 1, \cdots, N-1, \forall t = 1, \cdots, \tau: \quad \lim_{\beta \to 0} \hat{g}_{W^k}^{\mathrm{EP}}(\beta, t) = \hat{g}_{W^k}^{\mathrm{ID}}(t) \tag{84}$$

*Proof of Theorem A.4.* This proof follows the exact same methodology as that of Ernoult et al. [2019]. For self-containedness though and because of some subtle differences, we carry out here the derivation. We first define:

$$\Delta s_t^k := d_\beta s_{t+1}^k|_{\beta=0} - d_\beta s_t^k|_{\beta=0}. \tag{85}$$

Note that since $s_{\beta,t=0}^k = s_\star$, $d_\beta s_t^k|_{\beta=0} = 0$ since $s_\star$ does not depend on $\theta$. Furthermore, note that by differentiating the equation satisfied by $s_{\beta,t+1}^k$ with respect to $\beta$ and evaluating the resulting expressions at $\beta = 0$ yields:

$$d_\beta s_{\beta,t+1}^k|_{\beta=0} = \partial_1 K(s_\star^k, W^k, x^k = s_\star^{k-1}) \cdot d_\beta s_{\beta,t}^k|_{\beta=0} - \delta s^k \tag{86}$$

In particular, evaluating Eq. (86) for $t = 0$ yields:

$$\Delta s_0^k = d_\beta s_{\beta,1}^k|_{\beta=0} - \underbrace{d_\beta s_{\beta,0}^k|_{\beta=0}}_{=0} = -\delta s^k. \tag{87}$$

Therefore, substracting Eq. (86) across two timesteps yields altogether:

$$\begin{aligned} \Delta s_t^k &= \partial_1 K(s_\star^k, W^k, x^k = s_\star^{k-1}) \cdot \Delta s_{t-1}^k \\ &= \nabla_1^2 \Phi(s_\star^k, W^k, x^k = s_\star^{k-1}) \cdot \Delta s_{t-1}^k \\ &= \nabla_1^2 \Phi(s_\star^k, W^k, x^k = s_\star^{k-1})^\top \cdot \Delta s_{t-1}^k \\ &= \partial_1 K(s_\star^k, W^k, x^k = s_\star^{k-1})^\top \cdot \Delta s_{t-1}^k \end{aligned} \tag{88}$$

Note that $\hat{g}_{W^k}^{\mathrm{EP}}(t)$ rewrites:

$$
\begin{aligned}
\hat{g}_{W^k}^{\mathrm{EP}}(\beta, t) &= -d_\beta \left( \nabla_2 \Phi(s_{\beta,t+1}^k, W^k, x^k = s_\star^{k-1}) \right) + \mathcal{O}(\beta^2) \\
&= -\nabla_{1,2}^2 \Phi(s_\star^k, W^k, x^k = s_\star^{k-1}) \cdot d_\beta s_{\beta,t+1}^k|_{\beta=0} + \mathcal{O}(\beta^2) \\
&= -\nabla_{1,2}^2 \Phi(s_\star^k, W^k, x^k = s_\star^{k-1}) \cdot \Delta s_t^k - \nabla_{1,2}^2 \Phi(s_\star^k, W^k, x^k = s_\star^{k-1}) \cdot d_\beta s_{\beta,t}^k|_{\beta=0} + \mathcal{O}(\beta^2) \\
&= -\underbrace{\nabla_{1,2}^2 \Phi(s_\star^k, W^k, x^k = s_\star^{k-1})}_{= \nabla_{2,1}^2 \Phi(s_\star^k, W^k, x^k = s_\star^{k-1})^\top} \cdot \Delta s_t^k + \hat{g}_{W^k}^{\mathrm{EP}}(\beta, t-1) + \mathcal{O}(\beta^2) \\
&= \partial_2 K(s_\star^k, W^k, x^k = s_\star^{k-1})^\top \cdot \left( -\Delta s_t^k \right) + \hat{g}_{W^k}^{\mathrm{EP}}(\beta, t-1) + \mathcal{O}(\beta^2) \qquad (89)
\end{aligned}
$$

Likewise, we can show that:

$$
\Delta x_{\beta,t}^k = \partial_3 K(s_\star^k, W^k, x^k = s_\star^{k-1})^\top \cdot \left( -\Delta s_t^k \right) + \Delta x_{\beta,t-1}^k + \mathcal{O}(\beta^2) \qquad (90)
$$

Altogether, Eq. (87), Eq. (88) Eq. (89) and Eq. (90) yield, denoting $\hat{g}_{W^k}^{\mathrm{EP}}(t) := \lim_{\beta \to 0} \hat{g}_{W^k}^{\mathrm{EP}}(\beta, t)$ and $\Delta x_t^k := \lim_{\beta \to 0} \Delta x_{\beta,t}^k$:

$$
\begin{aligned}
&\forall k = N-1, \cdots, 1 : \\
&-\Delta s_0^k = \delta s^k = \Delta x_\tau^{k+1} \text{ if } k < N-1 \text{ else } \nabla_1 \tilde{\ell}(s_\tau^{N-1}, y), \ \hat{g}_{W^k}^{\mathrm{EP}}(0) = 0, \ \Delta x_0^k = 0 \qquad (91) \\
&\forall t = 1, \cdots, \tau : \\
&\left\{
\begin{array}{ll}
-\Delta s_t^k &= \partial_1 K(s_\star^k, W^k, x^k = s_\star^{k-1})^\top \cdot (-\Delta s_{t-1}^k) \\
\hat{g}_{W^k}^{\mathrm{EP}}(t) &= \hat{g}_{W^k}^{\mathrm{EP}}(t-1) + \partial_2 K(s_\star^k, W^k, x^k = s_\star^{k-1})^\top \cdot \left( -\Delta s_t^k \right) \\
\Delta x_t^k &= \partial_3 K(s_\star^k, W^k, x^k = s_\star^{k-1})^\top \cdot \left( -\Delta s_t^k \right) + \Delta x_{t-1}^k
\end{array}
\right. \qquad (92)
\end{aligned}
$$

Starting from $k = N-1$, $(-\Delta s_t^{N-1})_{t \in [\![1,\tau]\!]}$ and $(\delta s_t^{N-1})_{t \in [\![1,\tau]\!]}$, $(\Delta x_t^{N-1})_{t \in [\![1,\tau]\!]}$ and $(\delta x^{N-1})_{t \in [\![1,\tau]\!]}$, $(\hat{g}_{W^{N-1}}^{\mathrm{EP}}(t))_{t \in [\![1,\tau]\!]}$ and $(\hat{g}_{W^{N-1}}^{\mathrm{ID}}(t))_{t \in [\![1,\tau]\!]}$ satisfy the same initial conditions and recursive equations, therefore there are all (pair-wise) equal for $t = 1, \cdots, \tau$. Therefore in particular, $\Delta x_\tau^{N-1} = \delta x^{N-1}$ such that $(-\Delta s_t^{N-2})_{t \in [\![1,\tau]\!]}$ and $(\delta s_t^{N-2})_{t \in [\![1,\tau]\!]}$ from the previous $(N-2)^{\text{th}}$ block satisfy the same initial conditions, such that the reasonning applying to $k = N-1$ recurses for $k < N-1$, which yields Eq. (84).

$\square$

### A.4.4 Details about Fig. 5

**Precise hyperparameters to reproduce Fig. 5 can be found inside our repository**. Fig. 7 precisely depict the architecture at use for these experiments.

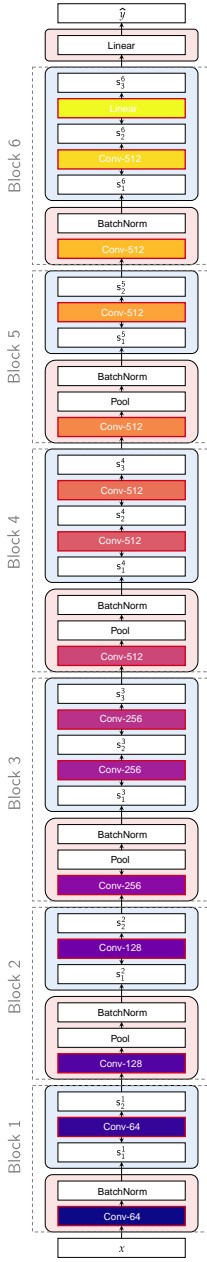

Figure 7: Architecture used for the static gradient analysis. The color code used to label layers matches that of Fig. 5. In the context of the static gradient analysis, "block" $k$ is defined as all layers participating in $\tilde{E}^k$, which therefore includes $F^k$ *and* $E^k$ modules (rather than one of these taken alone).

## A.5 Experimental Details

### A.5.1 Datasets

Simulations were run on CIFAR-10, CIFAR-100 and Imagenet32 datasets, all consisting of color images of size $32 \times 32$ pixels. CIFAR-10 [Krizhevsky, 2009] includes 60,000 color images of objects and animals. Images are split into 10 classes, with 6,000 images per class. Training data and test data include 50,000 images, and 10,000 images respectively. CIFAR-100 [Krizhevsky, 2009] likewise comprises 60,000 and features a diverse set of objects and animals split into 100 distinct classes. Each class contains 600 images. Like CIFAR-10, the dataset is divided into a training set with 50,000 images and a test set containing the remaining 10,000 images. The ImageNet32 dataset [Chrabaszcz et al., 2017] is a downsampled version of the original ImageNet dataset Russakovsky et al. [2015] containing 1,000 classes with 1,281,167 training images, 50,000 validation images, 100,000 test images and 1000 classes.

### A.5.2 Data preprocessing

All data were normalized according to statistics shown in 2 and augmented with 50% random horizontal flips. Images were randomly cropped and padded with the last value along the edge of the image.

Table 2: Data Normalization. Input images were normalized by conventional mean ($\mu$) and standard deviation ($\sigma$) values for each dataset. All images used are color (three channels).

| Dataset | Mean ($\mu$) | Standard deviation ($\sigma$) |
|---|---|---|
| CIFAR-10/100 | (0.4914, 0.4822, 0.4465) | (0.2470, 0.2435, 0.2616) |
| Imagenet32 | (0.485, 0.456, 0.406) | (0.3435, 0.336, 0.3375) |

### A.5.3 Simulation details

**Weight initialization.** EP, similar to other machine learning paradigms reliant on fixed-point iteration [Bai et al., 2019], is highly sensitive to initialization statistics [Agarwala and Schoenholz, 2022], hence conventionally difficult to tune, and requiring many iterations for the three relaxation phases. Initialization of weights as *Gaussian Orthogonal Ensembles* (GOE) ensures better stability (reduced variance) and, combined with other stabilizing measures discussed below, empirically yields faster convergence.
According to GOE, weights are intialized as:

$$W_{ij} \sim \begin{cases} \mathcal{N}(0, \frac{V}{N}), & \text{if } i \neq j \\ \mathcal{N}(0, \frac{2V}{N}), & \text{if } i = j \end{cases}$$

where $\mathcal{N}(\mu, \sigma^2)$ denotes a Gaussian (normal) distribution with mean $\mu$ and variance $\sigma^2$. $N$ was manually tuned for each architecture.

**State initialization.** All layers are initialized as zero matrices.

**Activation functions.** An important detail for faithful reproduction of these experiments is the choice and placement of activation functions applied during the iterative fixed-point procedure. In the literature, activations (i.e. "clamping") is conventionally applied at each layer, *with the exception of the final layer*, where it is sometimes included e.g. Scellier et al. [2024], and sometimes omitted Laborieux et al. [2021], depending on the loss function at use. For these experiments we used both the standard hard activation employed by Ernoult et al. [2019] and Scellier et al. [2024], and the more conservative one given in [Laborieux et al., 2021]. For the tolerance based and splitting experiments, we generalize the approach of Laborieux et al. [2021], by scaling values by a variable factor $\alpha$ instead of a fixed value $0.5$ . Details are given in Table 3.

In practice, we find that the smaller scaling factors corresponding with the "laborieux" activation, in conjunction with GOE, and the omission of clamping at the output of *each block* significantly

Table 3: Activation functions

| Name | Description | Source |
|------|-------------|--------|
| ernoult | $\sigma(x) = \max(\min(x, 1), 0)$ | [Ernoult et al., 2019] |
| laborieux | $\sigma(x) = \max(\min(0.5 \times x, 1), 0)$ | [Laborieux et al., 2021] |
| nest | $\sigma(x) = \max(\min(\alpha \times x, 1), 0)$ | This work |

enhances gradient stability and speeds convergence in deep multi-block settings. In the interest of multi-scale uniformity and consistency with previous literature [Laborieux et al., 2021] Ernoult et al. [2019], we apply clamping activations on *all layers* in our 6-layer architecture.

For the scaling experiments, we apply the "laborieux" activation at every layer *except* the output of each block. For the 12-layer splitting experiment, we do the same, omitting clamping from the output of the final layer of *each* block in the block-size=4 and block-size=3 experiments. However, in the block-size=2 case we clamp the output of the second and fourth blocks to preserve dynamics of the block-size=4 split. Such consistency is not possible for the block-size=3 experiment, constituting a possible discrepancy in dynamics.

**Cross-entropy loss and softmax readout.**   Following [Laborieux et al., 2021], all models were implemented such that the output $y$ is removed from the system (e.g. not included in the relaxation dynamics) but is instead the function of a weight matrix: $W_{\text{out}}$ of size $\dim(y) \times \dim(s)$, where $s$ is the state of the final layer. For each time-step $t$, $\hat{y}_t = \text{softmax}(W_{\text{out}} s_t)$.

The cross-entropy cost function associated with the softmax readout is then:

$$l(s, y, W_{\text{out}}) = -\sum_{c=1}^{C} y_c \log(\text{softmax}_c(W_{\text{out}} \cdot s)).$$

**Convention to count layers.**   It is important to note that by convention we refer to architectures throughout this text to the exclusion of the softmax readout, which is technically an additional layer, though not involved in the relaxation process.

**Architecture.**   All convolutional layers used in experiments are of kernel size 3 and stride and padding 1. Max-pooling was applied with a window of $2 \times 2$ and stride of 2. For the 6-layer model used in Table **??** , batchnorm was applied *after* the first layer convolution and pooling operation. All other models in both experiments use batch-normalization on the first layer of each block *after* convolution and pooling (where applied). These details exclude the linear softmax readout of size $n$ classes.

**Hyperparameters.**   **Detailed hyperparameters for to reproduce Table ?? and Table 1 are given inside the configuration files of our repository**. Note that all architectural details for the 12-layer models are *identical* across splitting and scaling experiments. Additionally, identical hyperparameters were used for CIFAR100 and Imagenet experiments of Table 1. Unlike previous literature, the use of GOE intialization eliminates the need for separate layerwise learning rates and initialization parameters. One noteworthy detail is that only 100 epochs were used for the larger model for Table 1 compared with 200 epochs for the smaller 12-layer model. This was due to prohibitively long run-time of training the larger model. Notably, performance still significantly improves with decreased overall runtime.

**Root-finding algorithms.**   While in principle any root-finding algorithm may be used for the two relaxation phases of our EP implementation (for inference and gradient computation), our implementation utilized a simple fixed-point iteration procedure, in which neuron states are initialized as zero vectors with values updated asynchronously on each iteration to that of the gradient of the total system energy with respect to current state. An approximate illustration of this procedure is found in Alg. 3. As indicated in Section 4.2, two variants of the convergence procedure were employed, one in which the average value of current state is compared to that of the previous state for each layer, and relaxation is truncated when values for all layers have a difference of *less than* 1e-4. This was known

as the tolerance-based (TOL) procedure. Notably, tolerance-based convergence criteria were applied *on the free phase only*, with nudging computed *with a fixed value of iterations*. This was to ensure consistency between ID and EP, though in practice a tolerance can be applied equally to the nudging phase.

---

**Algorithm 13** `Asynchronous with Tolerance` (for all blocks **until penultimate**)

---

*Input*: $T, \theta^k, \omega^k, s_\star^{k-1}, \beta, \delta s^k$
*Output*: $s_\beta^k$

1: $s^k \leftarrow 0$
2: $c \leftarrow \infty$
3: **for** $t = 1 \cdots T$ **do**
4:     $\forall$ odd $\ell \in \{1, \cdots, L_k\}: s_{\ell,\beta,temp}^k \leftarrow \sigma\left(\nabla_{s_\ell^k}\Phi\left(s_\beta^k, \theta^k, s_\star^{k-1}, \omega^k\right) - \beta\delta s^k\right)$
5:     $\forall$ even $\ell \in \{1, \cdots, L_k\}: s_{\ell,\beta,temp}^k \leftarrow \sigma\left(\nabla_{s_\ell^k}\Phi\left(s_\beta^k, \theta^k, s_\star^k, \omega^k\right) - \beta\delta s^k\right)$
6:     **if** $t \geq 2$ **then**
7:         $\forall \ell \in \{1, \cdots, L_k\}: c_\ell^k \leftarrow \frac{s_{\ell,\beta,temp}^k - s_{\ell,\beta}^k}{|s_{\ell,\beta}^k|}$
8:         **if** $\text{mean}(c^k) \leq Tol$ **then**
9:             **BREAK;**
10:         **end if**
11:     **end if**
12:     $s_{\ell,\beta}^k \leftarrow s_{\ell,\beta,temp}^k$
13: **end for**

---

**Supplementary results with a fixed number of iterations.** In addition to the TOL-based procedure, we obtained results for 4.2 using the more conventional approach of [Scellier and Bengio, 2019][Laborieux et al., 2021][Ernoult et al., 2019], applying fixed number of iterations on the first and second relaxation phases (see **??**). This approach was also the default used for our scaling experiments in 4.3. Importantly, with the TOL procedure described above Alg 3 becomes Alg 13. Results using a fixed iteration root-finding scheme are shown in 4

Table 4: Validation accuracy and Wall Clock Time (WCT) obtained on CIFAR-10 by EP (Alg. 2) and ID on models with different number of layers ($L$) and block sizes ("bs"). 3 seeds are used.

| | EP | | ID | |
| | Top-1 (%) | WCT | Top-1 (%) | WCT |
|---|---|---|---|---|
| **L =6** | | | | |
| bs=6 | $88.8^{\pm 0.2}$ | 8:06 | $87.3^{\pm 0.6}$ | 8:05 |
| bs=3 | $89.5^{\pm 0.2}$ | 8:01 | $89.2^{\pm 0.2}$ | 7:40 |
| bs=2 | $90.1^{\pm 0.2}$ | 7:47 | $90.0^{\pm 0.2}$ | 7:18 |
| **L =12** | | | | |
| bs=4 | $91.6^{\pm 0.1}$ | 7:49 | $91.6^{\pm 0.1}$ | 7:08 |
| bs=3 | $92.2^{\pm 0.2}$ | 6:06 | $92.2^{\pm 0.1}$ | 5:59 |
| bs=2 | $91.7^{\pm 0.2}$ | 6:10 | $91.8^{\pm 0.1}$ | 6:08 |

**Other details.** All experiments were run using Adam optimizer [Kingma and Ba, 2014]and Cosine Annealing scheduler[Loshchilov and Hutter, 2017], specifying some minimum learning rates and setting maximum T equal to epochs (i.e. no warm restarts). Code was implemented in Pytorch 2.0 and all simulations were run on NVIDIA A100 SXM4 40GB GPUs.

