# OpenReview forum: "Casting hybrid digital-analog training into hierarchical energy-based learning"
_NeurIPS.cc/2024/Workshop/MLNCP — MLNCP Oral_

### Official Review · Reviewer_9Ddy · 2024-09-19
**The paper introduces a way to combine feed-forward and energy-based blocks in models for neuromorphic computing**

**Rating:** 8
**Confidence:** 4

**Review:**

This paper derives expressions and algorithms for performing inference and weight update on models of deep networks that contain alternating blocks of feed forward layers and energy-based models. This was previously hard to do because, while feed forward layers are simple functions, energy-based models require internal optimization and an efficient algorithm to train (like Equilibrium Propagation).

This is a nice paper, though I am not completely convinced by its motivation. Since this combining of feed-forward and energy-based components, both backprop and equilibrium propagation are required, it is unclear what benefit is computationally obtained by this combination. EP and its variants make sense for physical analog implementations precisely because only such physical local rules work for them. I can imagine combining physical self-learning elements using EP with computational models that use BP, but the advantage for doing that is not apriori clear. The authors partially address this by noting that you can break down the model into ‘independent’ EB blocks that can significantly reduce computations time, as shown in Table 3. This should be communicated in the introduction, and possibly also in the abstract to help motivate the work.

I find this paper interesting and definitely meriting acceptance to the MLNCP workshop. Some minor comments are appended below:
1) Why is EP incompatible with common non-stationary operations such as activation functions. Is this because such functions cannot be simply written as “energy-based” elements? There are most likely ways of fixing this in the majority of cases.
2) Eq. 2 is noted incorrectly in the text. It probably should refer to the minimization of C.
3) What does “L” in EBL stand for?
3) Indexing of F,w on ln 93 may be incorrect. Didn’t you mean F^2,w^2?
4) In Figure 2, given the y-scale from 0-1 there seems to be no difference and all layers do equally well. It might be more informative to show log⁡(1-cos⁡(g^EP - g^AD ) ).

---

### Official Review · Reviewer_dcYF · 2024-10-02
**Interesting Hybrid Approach in Simulation**

**Rating:** 7
**Confidence:** 3

**Review:**

This paper describes an algorithm for training mixed feed forward (FF) and energy based (EB) networks. The methods are clear, and the results are encouraging. The observation that implementations of EB algorithms are far from mature, and therefore there is room for hybrid approaches, is a good one.

A few minor comments and questions:

Do the authors have any way of estimating the type of energy savings such algorithms could provide? That seems to be a main motivation of EB algorithms, and presumably the savings cannot be higher than the fraction of the network that is EB instead of FF. Or is there another motivation for including EB that is less straightforward?

The reference to Eq (2) should be to the second (unnumbered) equation, I believe.

The scale of Fig 2 makes it impossible to tell if there is any variation between layers – could the authors rescale or plot on a log scale 1- similarity?

The authors hint at but do not explicitly confirm that the advantage their system has over purely EB methods are nonlocal operations like batchnorm – is this the source of improved performace?

---

### Decision · Program_Chairs · 2024-10-10

Accept (Oral)